

# Solving 3d gravity with Virasoro TQFT

**Scott Collier[1]⋆, Lorenz Eberhardt[2]† and Mengyang Zhang[3]‡**

**1** Princeton Center for Theoretical Science, Princeton University, Princeton, NJ 08544, USA
**2** Institute for Advanced Study, Einstein Drive, Princeton, NJ 08540, USA
**3** Joseph Henry Laboratories, Princeton University, Princeton, NJ 08544, USA

⋆ scott.collier@princeton.edu , † elorenz@ias.edu , ‡ mengyang@princeton.edu

## Abstract

We propose a precise reformulation of 3d quantum gravity with negative cosmological constant in terms of a topological quantum field theory based on the quantization of the Teichmüller space of Riemann surfaces that we refer to as "Virasoro TQFT". This TQFT is similar, but importantly not equivalent, to $SL(2, \mathbb{R})$ Chern-Simons theory. This sharpens the folklore that 3d gravity is related to $SL(2, \mathbb{R})$ Chern-Simons theory into a precise correspondence, and resolves some well-known issues with this lore at the quantum level. Our proposal is computationally very useful and provides a powerful tool for the further study of 3d gravity. In particular, we explain how together with standard TQFT surgery techniques this leads to a fully algorithmic procedure for the computation of the gravity partition function on a fixed topology exactly in the central charge. Mathematically, the relation leads to many nontrivial conjectures for hyperbolic 3-manifolds, Virasoro conformal blocks and crossing kernels.

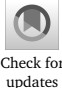

# 1  Introduction

In the study of quantum gravity, a central objective is the construction of tractable models that serve as theoretical laboratories where fundamental questions can be precisely posed. However the construction of soluble models of quantum gravity is notoriously difficult. This problem becomes much more tractable in low dimensions. In particular in three spacetime dimensions, pure Einstein gravity does not carry dynamical graviton degrees of freedom, i.e. there are no gravitational waves. This drastically simplifies the theory and there is hope to construct a consistent theory of quantum gravity from the bottom up without the need to resort to string theory. Moreover Einstein gravity with negative cosmological constant admits nontrivial black hole solutions [1], so despite the absence of propagating gravitons, the three-dimensional case retains much of the richness and many of the conceptual puzzles of the higher-dimensional problem. A full treatment of quantum gravity in two-dimensions in form of dilaton gravity models has been achieved, see e.g. the recent progress on JT gravity [2–4], but the status of the much richer three-dimensional case is still unclear. Three-dimensional gravity behaves in many ways as a topological field theory, where the role of edge modes is played by graviton excitations at the asymptotic boundaries. Since many three-dimensional topological field theories are solvable at the quantum level, this language provides the natural framework for an exact formulation of 3d quantum gravity.

This description can be made more concrete in the case of negative cosmological constant through the holographic principle. The AdS/CFT correspondence provides a window into the ultraviolet dynamics of pure 3d quantum gravity by reformulating the problem in terms of the boundary CFT. Indeed, the asymptotic symmetries of 3d gravity are given by two copies of the Virasoro algebra with central charges $c = \frac{3\ell}{2G_N}$ [5], where $\ell$ is the AdS radius, indicating the existence of a boundary CFT dual to 3d gravity. Despite much progress the status of the boundary CFT is still not fully settled, but there are many (partially competing) propos-

als [6–16]. Drawing on lessons learned in the recent solution of two-dimensional gravity [2], it has been conjectured that the boundary description is not given by a particular family of 2d CFTs admitting a large-$c$ limit, but rather by an appropriate notion of an ensemble of chaotic large-$c$ CFTs. In contrast to the case of two-dimensional gravity, the boundary description is subject to the tremendously rigid constraints of locality in the form of modular invariance and associativity of the operator product expansion (OPE), so the averaging cannot be done in an arbitrary way; indeed there is naively some tension between the rigidity of such consistency conditions and expectations from quantum chaos. While there has been partial progress on the construction of such an ensemble [14, 15], it suffers from the obvious problem that there is no known concrete example of a compact unitary large-$c$ CFT with a sparse light spectrum and only Virasoro symmetry[1] — our knowledge of the solutions to the crossing equations, particularly in the holographic regime, is embarrassingly limited. The existence of the boundary CFT description together with the data of the low-energy effective gravitational theory imposes powerful constraints on the UV degrees of freedom of the bulk theory in the form of conformal bootstrap techniques [18–28].

In this paper, we will address the problem from the bulk perspective. As a main technical result, we will make the relation of AdS$_3$-gravity with the TQFT description precise. This opens up an avenue for the computation of many bulk quantities that were of reach before.

We start by reviewing the canonical quantization of 3d gravity in terms of the quantization of Teichmüller space and then relate the gravity theory to a specific TQFT that we refer to as "Virasoro TQFT," which has previously appeared in the literature under various guises [29–34]. This formulation automatically provides a Hilbert space description for pure AdS$_3$ gravity, and allows us to use standard TQFT techniques to calculate the partition functions on various 3d manifolds. We will now give a bird's-eye view of logic of the paper.

The TQFT treatment of pure AdS$_3$ gravity has been proposed and studied for decades, starting from the connections between PSL$(2,\mathbb{R}) \times$ PSL$(2,\mathbb{R})$ Chern-Simons theory and Einstein gravity in AdS$_3$ spacetime at the classical level [35, 36]. Via a field redefinition, we can trade the dreibein and spin connection from the first-order formalism of 3d gravity with two $\mathfrak{sl}(2,\mathbb{R})$ gauge fields $A, \bar{A}$. The classical actions of two theories coincide. However, the direct quantization of PSL$(2,\mathbb{R}) \times$ PSL$(2,\mathbb{R})$ Chern-Simons theory is *not* equivalent to the quantization of pure 3d gravity. The phase spaces of the two theories are importantly different. Basic principles of the gravity theory require the metric to be non-degenerate, while Chern-Simons theory does not have such a requirement, so $\mathcal{M}_{\text{gravity}} \subsetneq \mathcal{M}_{\text{CS}}$; in the Chern-Simons description one would be integrating over gauge field configurations that are non-sensical when interpreted as three-dimensional metrics. The gravity phase space $\mathcal{M}_{\text{gravity}}$ on a spatial slice $\Sigma$ is in fact equal to two copies of the Teichmüller space $\mathcal{T}$ of $\Sigma$ [37–39]. Teichmüller space is the universal covering space of the moduli space of Riemann surfaces and thus roughly describes the shape of the initial value surface.

It is a non-trivial (and from the perspective of the gauge theory, somewhat miraculous) fact that Teichmüller space can be consistently quantized on its own [29,30,33,40–44]. This means that we can assign to $\mathcal{T}$ a quantum Hilbert space $\mathcal{H}_\Sigma$ endowed with a well-defined inner product. However, in contrast to more standard TQFTs, this Hilbert space is infinite-dimensional. In physical terms, the Hilbert space is simply given by the space of all (normalizable) Virasoro conformal blocks on $\Sigma$. In fact, one can go further. Teichmüller space enjoys a natural action of the two-dimensional mapping class group, a.k.a. modular transformations or crossing transformations acting on $\Sigma$. These are represented as unitary operators on Hilbert space implementing the usual crossing transformations on conformal blocks. The remarkable fact that (normalizable) Virasoro conformal blocks close under crossing ensures independence of the

---

[1]See however [17] for recent progress constructing candidate examples of compact unitary CFTs with no chiral symmetry enhancements beyond Virasoro.

choice of basis for the conformal blocks in the quantization and means that one can define a consistent 3d TQFT from this data. This is a version of the Moore-Seiberg construction [45] generalized to the non-rational setting. This TQFT was previously formulated in the mathematics literature by Andersen and Kashaev [31,32]. However, we give a different treatment of it that is physically better motivated and which gives a much more convenient language for holography.

Therefore, we propose that in order to solve pure AdS$_3$ gravity, we should consider Virasoro TQFT instead of SL(2, $\mathbb{R}$) Chern-Simons theory. More precisely, we propose that the gravity partition function $Z_{\mathrm{grav}}(M)$ on a manifold $M$ with fixed topology can be written as follows in terms of the Virasoro TQFT partition function $Z_{\mathrm{Vir}}(M)$

$$Z_{\mathrm{grav}}(M) = \sum_{\gamma \in \mathrm{Map}(\partial M)/\mathrm{Map}(M, \partial M)} |Z_{\mathrm{Vir}}(M^{\gamma})|^2. \qquad (1.1)$$

This equation holds whenever $M$ is a hyperbolic 3-manifold with at least one AdS boundary. There is a modification in the absence of asymptotic boundaries (2.66) and a version that partially holds in the non-hyperbolic case (2.63) that we shall explain in section 2.7.

We now briefly explain the ingredients going into eq. (1.1).

**Virasoro TQFT partition function.** The equation (1.1) involves two copies of the Virasoro TQFT partition function $Z_{\mathrm{Vir}}(M)$ on the manifold $M$. One copy should be thought of as being left-moving, the other as being right-moving. We emphasize that the partition function of Virasoro TQFT is algorithmically exactly computable on hyperbolic 3-manifolds. Computability is achieved by standard surgery techniques familiar from Chern-Simons theory [46]. The relation of Virasoro TQFT to Liouville CFT is exactly analogous to the relation of Chern-Simons theory to the WZW model. In particular, as already mentioned above, the Hilbert space of Virasoro TQFT consists of the space of all Virasoro conformal blocks on a spatial surface $\Sigma$. The space of conformal blocks is equipped with an interesting inner product given in terms of an integral over Teichmüller space, see eq. (2.5) in the main text. We however argue from a variety of perspectives that the required integral can be computed explicitly in terms of the DOZZ structure constants [47,48] of Liouville theory. The Hilbert space also carries the action of all crossing transformations which in particular defines a projective, unitary representation of the mapping class group $\mathrm{Map}(\Sigma_g)$. This representation is known very explicitly thanks to the pioneering work of Ponsot and Teschner [43,49–51].

As we shall review in detail, this structure is enough to compute partition functions of the three-dimensional theory using surgery techniques. However, compared to e.g. SU(2) Chern-Simons theory, some extra care is needed. Since the Hilbert space of the theory is not well-defined on the sphere with fewer than three insertions and on the torus in the absence of insertions, we get ill-defined results if we try to perform surgery of manifolds along such surfaces. Additionally, since the Hilbert space is infinite-dimensional, inner products are not always guaranteed to be finite. Thus the Virasoro TQFT partition function does not take finite values on all three-dimensional manifolds. Moreover, similarly to Chern-Simons theory, $Z_{\mathrm{Vir}}(M)$ has a framing anomaly, which however cancels once left- and right-mover are combined in (1.1).

We will explain a concrete algorithm to compute these partition functions by using a (generalized) Heegaard splitting of the manifold under consideration. This procedure gives a finite result whenever $Z_{\mathrm{Vir}}(M)$ is well-defined. In a follow up paper [52], we demonstrate the effectiveness of this algorithm by computing explicitly the TQFT partition function for various known concrete examples of hyperbolic 3-manifolds, such as the figure-8 knot complement. For standard examples in the literature, this completely trivializes the computation. We also

calculate the partition functions for several multi-boundary wormhole geometries, which encode the non-Gaussian statistics of holographic CFT data [14, 15, 53].

**Mapping class group.** A theory of gravity obviously cannot be equivalent to a TQFT. In a TQFT the background manifold is given, whereas it is dynamical in a gravity theory. In particular, there are large diffeomorphisms in a gravity theory that are not gauge transformations from the TQFT point of view. They are captured by the mapping class groups in the problem. There is a bulk and a boundary mapping class group,

$$\text{Map}(M, \partial M) \equiv \text{Diff}(M, \partial M)/\text{Diff}_0(M, \partial M), \quad \text{Map}(\partial M) \equiv \text{Diff}(\partial M)/\text{Diff}_0(\partial M). \quad (1.2)$$

Here, $\text{Diff}(M, \partial M)$ are diffeomorphisms of the bulk manifold that are allowed to act nontrivially on the boundary of the manifold (but preserve each boundary component setwise). The mapping class group is responsible for that fact that quantum gravity does not follow basic QFT axioms such as factorization of amplitudes.

The sum over topologies in the quantum gravity path integral includes the sum over the boundary mapping class group, while we still have to gauge by the bulk mapping class group. However, when $M$ has boundaries, the bulk mapping class group $\text{Map}(M, \partial M)$ naturally acts on the boundary. For a hyperbolic 3-manifold $M$, one can show that the map $\text{Map}(M, \partial M) \longrightarrow \text{Map}(\partial M)$ has no kernel and we can view $\text{Map}(M, \partial M) \subset \text{Map}(\partial M)$. This is a relatively deep geometric fact which we explain it in Section 2.7 and Appendix C. This is in very stark contrast with the two-dimensional case. Thus, the sum over topologies and the gauging of the bulk mapping class group partially cancel which leads to eq. (1.1). In the formula $M^\gamma$ denotes the image of $M$ under the action of the mapping class group element $\gamma$. For simple examples such as the sum over modular images of the BTZ black hole, this recovers the well-known modular sum as first studied in [9].

Let us make some further remarks. It is perhaps surprising that the left- and right-movers factorize as nicely as in eq. (1.1). The only 'entanglement' between them comes from the modular sum over the mapping class group. This sum in particular ensures that all the spins as measured on the boundary only take integer values. Of course, (1.1) still needs to be summed over different topologies to obtain the partition function of full 3d quantum gravity. Apart from some new observations in the discussion section 4, we do not analyze this sum in this paper, but rather focus on the contribution from a fixed topology. Similarly, our paper does not immediately lead to conceptually new predictions about the boundary CFT description, but serves as a useful tool to further analyze a possible boundary description. We will explore the implications for the boundary CFT further in a follow-up paper [52]. We should also mention that the prescription (1.1) gives the answer for the partition function for any hyperbolic manifold, but the general prescription for 'off-shell' partition functions is still open. In that case, both $Z_{\text{Vir}}(M)$ can be divergent and there can be an infinite bulk mapping class group by which we need to divide [54]. We comment further on these cases in the discussion section 4.

Finally, we want to emphasize that there is a number of seemingly lucky coincidences that are essential for this correspondence to work, which we believe also serves as evidence for its correctness. For instance, it is very fortunate that the phase space is given in terms of two Teichmüller spaces which admit a natural quantization in terms of Virasoro conformal blocks. In general, restricting the phase space in an arbitrary manner does not lead to a well-defined theory. Moreover, three-dimensional hyperbolic manifolds behave very differently from two-dimensional hyperbolic manifolds, which leads to a simple treatment of the mapping class group, at least for the hyperbolic case. In comparison, one needs the Mirzakhani recursion relations to solve the two-dimensional case [55, 56]. So the surgery technique of the Virasoro

TQFT may heuristically be understood as a three-dimensional version of Mirzakhani's recursion relation.[2]

This paper is organized as follows. In section 2, we review the construction of the gravity phase space of pure AdS$_3$ gravity and its identification with Teichmüller space. We review the quantization of Teichmüller space and the construction of the Virasoro TQFT. This in particular includes a detailed discussion of the inner product of the Hilbert space. We then derive eq. (1.1) in detail. We also explain the parallel logic for the 2d analogue of JT gravity. In section 3, we explain how to algorithmically compute the partition function of the Virasoro TQFT. We explain the trivial computation of the partition function of the Euclidean wormhole in our framework and then explain computations via surface bundles and (generalized) Heegaard splittings. We provide three appendices for supplemental material. In appendix A, we list the consistency consitions satisfied by the crossing kernels. In appendix B, we explain our conventions for both spacelike and timelike Liouville theory, which both play an important role at various places in the main text. Finally, we explain various standard facts about hyperbolic 3-manifolds in appendix C.

**Note.** The TQFT we consider in this paper has sometimes been referred to as "Teichmüller TQFT" in the literature. We prefer the name "Virasoro TQFT" because we feel it better captures the physical meaning of the TQFT, and we strongly feel that a person like O. Teichmüller does not deserve additional scientific honor for a recent development that he did not contribute to. Moreover our treatment of the theory is sufficiently different from previous considerations that its equivalence to the theory known as Teichmüller TQFT, particularly the restricted state-sum construction of [31, 32], while expected, remains to be demonstrated.

# 2 3d gravity and its relation to Virasoro TQFT

## 2.1 3d gravity in first order formalism

It is well-known that 3d gravity is related to $\mathrm{PSL}(2, \mathbb{R}) \times \mathrm{PSL}(2, \mathbb{R})$ Chern-Simons theory [35, 36]. This correspondence states that 3d gravity in first order formalism admits a field redefinition that maps the equation of motions and the action to those of $\mathrm{SL}(2, \mathbb{R})_k \times \mathrm{SL}(2, \mathbb{R})_{-k}$ Chern-Simons theory (up to global issues mentioned below). In terms of the dreibein $e_\mu^a$ and the dualized spin connection $\omega_\mu^a = \frac{1}{2} \varepsilon^{abc} \omega_{\mu,bc}$, the relevant change of variables is

$$A_\mu^a = \omega_\mu^a + \frac{1}{\ell} e_\mu^a, \qquad \bar{A}_\mu^a = \omega_\mu^a - \frac{1}{\ell} e_\mu^a, \tag{2.1}$$

where $A_\mu$ and $\bar{A}_\mu$ become the $\mathfrak{sl}(2, \mathbb{R}) \times \mathfrak{sl}(2, \mathbb{R})$ gauge fields and $\Lambda = -\ell^{-2} < 0$ is the cosmological constant. The level $k$ of the Chern-Simons theory is related to the AdS-length $\ell$ and Newton's constant as $k = \frac{\ell}{4G}$, which is classically related to the Brown-Henneaux central charge $c$ as $c = 6k$ [5].

While this identification works formally on the level of the classical fields, one is immediately faced with myriad problems in its interpretation:

1. In 3d gravity, the metric is a dynamical field and as such the path integral includes a sum over different topologies. On the contrary, we consider a fixed background topological manifold for Chern-Simons theory and it is unnatural to sum over such choices.

---

[2]Although unlike Mirzakhani's recursion, the mapping class group must be treated separately.

2. In 3d gravity, there is a condition that the metric is Lorentzian (and in particular the dreibein is non-degenerate). In Chern-Simons theory, the gauge fields $A_\mu^a$ and $\bar{A}_\mu^a$ are not subject to such a non-degeneracy condition.

3. In 3d gravity, the gauge group is $\mathrm{Diff}(M)$ when we consider the theory on a manifold $M$.[3] This group agrees infinitesimally with gauge transformations from the Lie algebra $\mathfrak{sl}(2,\mathbb{R}) \times \mathfrak{sl}(2,\mathbb{R})$. However, the global structure is different. In particular, the group of diffeomorphisms is in general disconnected. Letting $\mathrm{Diff}_0(M)$ the identity component of the group of diffeomorphisms, the quotient group $\mathrm{Map}(M) = \mathrm{Diff}(M)/\mathrm{Diff}_0(M)$ is known as the mapping class. Chern-Simons theory can only capture the identity component $\mathrm{Diff}_0(M)$.

In this paper, we will propose a prescription that solves these problems for a large class of 3-manifolds – namely at least all hyperbolic 3-manifolds. We will start by explaining Virasoro TQFT – a theory closely related to 3d gravity that completely resolves the second problem, but still has the others, which we argue can be fixed by hand.

## 2.2 The phase space and the Teichmüller component

Let us consider a 3-manifold of the form $\Sigma \times \mathbb{R}$ for $\Sigma$ a Riemann surface. As usual, we can attach to $\Sigma$ the phase space describing initial conditions. In $\mathrm{SL}(2,\mathbb{R})$ Chern-Simons theory, the phase space is given by the moduli space of flat $\mathrm{SL}(2,\mathbb{R})$ bundles on $\Sigma$. However, not any flat $\mathrm{SL}(2,\mathbb{R}) \times \mathrm{SL}(2,\mathbb{R})$ bundle corresponds to a good initial condition for gravity since some bundles may look very singular in the metric variables (2.1). Thus one wants to impose a non-degeneracy condition on the space of flat bundles.

In 3d gravity with negative cosmological constant, this condition turns out to be quite natural and singles out the Teichmüller component in the moduli space of flat $\mathrm{SL}(2,\mathbb{R})$ bundles [37,38]. Recall that flat $\mathrm{SL}(2,\mathbb{R})$ bundles (or rather flat $\mathrm{PSL}(2,\mathbb{R})$) bundles are classified topologically by their Euler number, which takes values in the range $\{-|\chi(\Sigma)|, \dots, |\chi(\Sigma)|\}$ (in the absence of conical defects). Teichmüller space is the component with maximal Euler number (or, equivalently after a orientation-reversal, minimal Euler number). It captures precisely the flat bundles that come from hyperbolic structures on $\Sigma$.

Thus the moduli space of flat $\mathrm{PSL}(2,\mathbb{R})$-bundles is disconnected and gravity only picks out the geometric component $\mathcal{T} \subset \mathcal{M}_{\mathrm{PSL}(2,\mathbb{R})}^{\mathrm{flat}}$. Hence we take the phase space of gravity to be two copies of Teichmüller space of the initial value surface $\Sigma$:

$$\text{Gravity phase space} = \mathcal{T} \times \overline{\mathcal{T}}. \tag{2.2}$$

We should think of one copy as left-moving and the other as right-moving.

Note that there is also an alternative description of phase space in terms of the cotangent bundle of Teichmüller space, which naturally is obtained from the Hamiltonian formalism of 3d gravity. They are symplectomorphic, $\mathcal{T} \times \overline{\mathcal{T}} \cong T^*\mathcal{T}$ [38,57]. For our purposes it will be more convenient to work with the factorized form. We also remark that the phase space is sometimes claimed to be $(\mathcal{T} \times \overline{\mathcal{T}})/\mathrm{Map}(\Sigma)$, where $\mathrm{Map}(\Sigma)$ is the 2d mapping class group on the initial value surface [39,54,58] (i.e. the group of modular transformations). In gravity, we can implement the mapping class group either before or after quantization. In practice, this means that modding by the 2d mapping class group is useful whenever the 2d mapping class group embeds into the 3d mapping class group. This happens for example for manifolds

---

[3]There is also a choice involved regarding whether we allow for orientation reversing diffeomorphisms or not. We will assume throughout this paper that $M$ is orientable, and $\mathrm{Diff}(M)$ is the group of orientation-preserving diffeomorphisms. This is the choice that corresponds most naturally to Chern-Simons theory, since CS theory is formulated on oriented 3-manifolds. The theory then has time-reversal symmetry $T$, which we are not gauging.

of the form $\Sigma \times S^1$ or for the Euclidean wormhole that is topologically of the form $\Sigma \times \mathbb{R}$, but not in general. We will retain more computational control by dividing by the mapping class group after quantization.

## 2.3 The Hilbert space

As a next step, one quantizes the phase space of the theory to obtain the Hilbert space associated to the initial value surface $\Sigma$. This problem reduces to the quantization of Teichmüller space, which was essentially solved in [29] and later refined in [30, 40, 43, 44]. Our discussion here however emphasizes different aspects than in those works.

Let us first explain what it means to quantize Teichmüller space. As any phase space, Teichmüller space is a symplectic manifold. The symplectic form is given by the Weil-Petersson symplectic form. Quantization assigns a Hilbert space to a symplectic manifold. It also promotes certain distinguished functions to operators acting on that Hilbert space and their Poisson brackets to commutators. In general, there is no preferred recipe to perform quantization. It however so happens that Teichmüller space is a Kähler manifold in which case one can use geometric quantization to quantize. It is however far from obvious that this is the only possible quantization of Teichmüller space.

We will now describe the result of the quantization in high-level terms. Kähler quantization proceeds in two steps. First, we have to find a holomorphic line bundle $\mathscr{L}$ over $\mathcal{T}$ whose first Chern class is the symplectic form, $c_1(\mathscr{L}) = \omega$. Since Teichmüller space is simply-connected, this line bundle is unique. This step is often called prequantization. In a second step, one declares the Hilbert space to be holomorphic sections of this line bundle. Thus, every wavefunction is such a holomorphic section. It depends in particular only on the holomorphic coordinates, which is the analogue of the usual fact that the wave function should only depend on either positions or momenta, but not both.

There is a natural candidate for such holomorphic sections, namely Virasoro conformal blocks. They depend holomorphically on the moduli of the surface $\Sigma$. Since they are not yet crossing symmetric, they are objects defined on Teichmüller space, rather than moduli space. The conformal anomaly means that they are not *functions* on Teichmüller space, but rather sections of a holomorphic line bundle. Thus the Hilbert space obtained by quantizing Teichmüller space consists of Virasoro conformal blocks on the surface $\Sigma$. The central charge is related to the Chern-Simons level $k$ as follows,

$$c = 1 + 6Q^2, \qquad Q = b + b^{-1}, \qquad b = \frac{1}{\sqrt{k-2}}. \qquad (2.3)$$

This value can be found from Hamiltonian reduction of $\mathrm{SL}(2,\mathbb{R})_k$ conformal blocks via the $H_3^+$/Liouville correspondence [59], which agrees with the classical Brown-Henneaux value $6k$ up to quantum corrections [5]. We will mostly consider the regime $c \geq 25$ and can choose $b \in [0,1]$, so that the classical limit corresponds to $b \to 0$. It is however in principle sufficient to impose $c > 1$ for the construction described in this paper to exist.[4] As usual, we parameterize conformal weights in terms of the "Liouville momenta" $\alpha$ or $P$ as[5]

$$\Delta = \alpha(Q - \alpha) = \frac{c-1}{24} + P^2, \qquad \alpha = \frac{Q}{2} + iP. \qquad (2.4)$$

Finally, we need to explain how to turn this into a Hilbert space by exhibiting an inner product on the space of conformal blocks. In geometric quantization, such an inner product

---

[4]For $1 < \mathrm{Re}(c) < 25$ the description of the inner product that follows must be altered slightly, but we expect the conclusions to remain essentially unchanged. Formally, we can even take $c \in \mathbb{C} \setminus (-\infty, 1]$, but it is unclear what the construction means physically if $c$ is not real.

[5]In this paper we use $\Delta$ to refer to the conformal weight, *not* the total scaling dimension.

generally takes the form of an integral over the symplectic manifold under consideration. See e.g. [60] for the case of Chern-Simons theory with a compact gauge group. For the quantization of Teichmüller space, the inner product was worked out in [29]. It takes the form

$$\langle \mathcal{F}_1 | \mathcal{F}_2 \rangle = \int_{\mathcal{T}} Z_{\text{bc}} Z_{\text{timelike Liouville}} \overline{\mathcal{F}}_1 \mathcal{F}_2 \,, \tag{2.5}$$

for two conformal blocks $\mathcal{F}_1$ and $\mathcal{F}_2$ on $\Sigma$. A little reflection makes it plausible that something like this is the only well-defined formula we could have written down. Indeed, the ingredients are well-known from string theory. Integration over Teichmüller space is locally the same as integration over the moduli space of Riemann surfaces. In particular, we need an object of central charge 26 for the Weyl anomaly to cancel. Thus we need to multiply by some CFT partition function of conformal weight $26 - c \le 1$. The only natural CFT with central charge $\le 1$ is timelike Liouville theory.[6] Finally, we need $bc$ ghosts to define the measure for the integration over Teichmüller space. In the presence of punctures, we have to combine the conformal blocks with appropriate vertex operators in the timelike Liouville theory such that the total conformal weight is 1 and the states are physical in the string theory sense.

We now remind the reader about timelike Liouville theory whose meaning was elucidated in [61–63], which we review in appendix B. Let us only mention here a convenient parametrization of the central charge and conformal weights of the theory. The central charge in timelike Liouville CFT is often parameterized as

$$\hat{c} = 1 - 6\widehat{Q}^2, \quad \widehat{Q} = \hat{b}^{-1} - \hat{b} \,. \tag{2.6}$$

Hence to cancel the Weyl anomaly we must have

$$\hat{c} = 26 - c \quad \implies \quad \hat{b} = b \,. \tag{2.7}$$

Meanwhile, conformal weights in timelike Liouville are written in terms of the Liouville momenta $\widehat{P}$ as

$$\hat{\Delta} = \widehat{\alpha}(\widehat{Q} + \widehat{\alpha}), \quad \widehat{\alpha} = -\frac{\widehat{Q}}{2} + \widehat{P} \,. \tag{2.8}$$

Thus on the solution to the physical state conditions we have

$$\hat{\Delta} = 1 - \Delta \quad \implies \quad \widehat{P} = \pm i P \,. \tag{2.9}$$

We have suppressed a subtlety in the inner product (2.5). The Liouville CFT carries a label $\mu$ known as the Liouville cosmological constant in addition to the central charge. A priori, the value of the Liouville cosmological constant represents an ambiguity of the inner product in the quantum Teichmüller theory. In 3d gravity this ambiguity reflects the freedom to add covariant counterterms associated with each asymptotic boundary via holographic renormalization. In what follows we will mostly ignore this ambiguity, which amounts to choosing an arbitrary, but fixed value for $\mu$. We will comment more on holographic renormalization in section 2.7.

With the definition of the inner product (2.5) in place, we can analyze which conformal blocks are normalizable. Since we are dealing with a non-compact phase space, we can only expect states to be delta-function normalizable. Divergences in the definition (2.5) come from the boundaries of Teichmüller space for which a closed curve on the Riemann surface pinches. We can for example analyze the case of two four-point blocks. Then as the cross ratio $z$ tends to 0, the product of conformal blocks behaves as

$$z^{-\Delta_1 - \Delta_2 + \Delta} \bar{z}^{-\Delta_1 - \Delta_2 + \Delta'} \,, \tag{2.10}$$

---

[6] At the time of Verlinde's paper [29], the subtleties of timelike Liouville theory were not appreciated and we take eq. (2.5) as a more precise definition of his formula.

where $\Delta_1$ and $\Delta_2$ are the external conformal weights. They are part of the data specifying the punctured surface and thus coincide for both conformal blocks, while $\Delta$ and $\Delta'$ are the two (potentially different) internal conformal weights of the blocks. The timelike Liouville partition function behaves for $z \to 0$ as [62]

$$|z|^{-2(\hat{\Delta}_1 + \hat{\Delta}_2) + \frac{\hat{c}-1}{12}} |\log|z||^{-\frac{1}{2}} , \tag{2.11}$$

where $\hat{\Delta}_i = 1 - \Delta_i$ is the conformal weight of the associated timelike Liouville vertex operator and $\hat{c} = 26 - c$ the central charge. Thus the integrand (2.5) behaves for $z \to 0$ as

$$z^{\Delta - \frac{c-1}{24} - 1} \bar{z}^{\Delta' - \frac{c-1}{24} - 1} |\log|z||^{-\frac{1}{2}} . \tag{2.12}$$

The integral over $z$ has thus a chance of converging when both internal conformal weights satisfy

$$\Delta > \frac{c-1}{24} , \tag{2.13}$$

i.e. are above threshold.

We will shortly provide a more computationally useful perspective on this inner product, but we find it instructive to first work through an example where the inner product (2.5) can be computed explicitly. In particular, consider the inner product between conformal blocks associated with the three-point function $\langle \mathcal{O}_1 \mathcal{O}_2 \mathcal{O}_3 \rangle$ of local operators on the sphere. Of course, these conformal blocks are trivial (we may take them to be normalized to unity[7]), and there is no Teichmüller space to integrate over. Hence the only nontrivial ingredient in (2.5) is the sphere three-point function in the timelike Liouville CFT, which is given by the timelike DOZZ formula [61–66]

$$\langle \mathcal{F}_{0,3} | \mathcal{F}_{0,3} \rangle = \widehat{C}_{\text{TLL}}(\widehat{P}_1, \widehat{P}_2, \widehat{P}_3) , \tag{2.15}$$

where the $\widehat{P}_i$ indicate the Liouville momenta of the appropriate operators in timelike Liouville CFT; on the solution to the physical state conditions, they are given in terms of those of the $\mathcal{O}_i$ as in (2.9). As reviewed in appendix B, we adopt operator normalization conventions such that the timelike Liouville structure constants are given explicitly by

$$\widehat{C}_{\text{TLL}}(\widehat{P}_1, \widehat{P}_2, \widehat{P}_3) = \frac{1}{C_0(i\widehat{P}_1, i\widehat{P}_2, i\widehat{P}_3)} \bigg|_{b = \hat{b}} . \tag{2.16}$$

Here, $C_0$ is the universal formula for CFT structure constants given in equation (B.3) that is equivalent to the DOZZ formula [47,48] for the three-point function in spacelike Liouville CFT with a particular choice of operator normalization.[8] So the three-point functions in timelike Liouville CFT are given by the analytic continuation of the inverse of those in spacelike Liouville CFT. We reproduce the explicit form of $C_0$ here for convenience:

$$C_0(P_1, P_2, P_3) = \frac{\Gamma_b(2Q)\Gamma_b(\frac{Q}{2} \pm iP_1 \pm iP_2 \pm iP_3)}{\sqrt{2}\Gamma_b(Q)^3 \prod_{k=1}^{3} \Gamma_b(Q \pm 2iP_k)} . \tag{2.17}$$

---

[7]Of course, even the conformal block on a three-punctured sphere has ambiguous normalization due to the conformal anomaly. What we mean is that we can choose the standard normalization for the three-punctured sphere block on the plane,

$$\mathcal{F}_{0,3} = z_{21}^{-\Delta_1 - \Delta_2 + \Delta_3} z_{31}^{-\Delta_1 - \Delta_3 + \Delta_2} z_{32}^{-\Delta_2 - \Delta_3 + \Delta_1} . \tag{2.14}$$

[8]The precise relationship between the universal formula $C_0$ and the DOZZ formula as conventionally written is reviewed in appendix B.

We take the product over all choices of $\pm$ signs in the formula. Here, $\Gamma_b(z)$ is the so-called double Gamma function. It can be characterized as the unique function $\Gamma_b(z)$ that is meromorphic in $z$ on the complex plane and continuous in $b \in \mathbb{R}$ and satisfies the following shift equation

$$\Gamma_b(z + b) = \frac{\sqrt{2\pi} b^{bz-\frac{1}{2}}}{\Gamma(bz)} \Gamma_b(z),\tag{2.18}$$

as well as the same relation with $b \to b^{-1}$. Finally, the normalization is fixed by requiring that $\Gamma_b(\frac{Q}{2}) = 1$. An explicit representation which holds for $\operatorname{Re} z > 0$ is given by

$$\log \Gamma_b(z) = \int_0^\infty \frac{dt}{t} \left( \frac{e^{\frac{t}{2}(Q-2z)} - 1}{4 \sinh(\frac{bt}{2}) \sinh(\frac{t}{2b})} - \frac{1}{8}(Q-2z)^2 e^{-t} - \frac{Q-2z}{2t} \right).\tag{2.19}$$

We thus conclude that the inner product between three-point blocks in Virasoro TQFT is given by the inverse of the structure constants in spacelike Liouville CFT

$$\langle \mathcal{F}_{0,3} \,|\, \mathcal{F}_{0,3} \rangle = \frac{1}{C_0(P_1, P_2, P_3)}.\tag{2.20}$$

We will now argue that this is an example of a more general phenomenon. Indeed, we claim that[9]

$$\langle \mathcal{F}_{g,n}^{\mathcal{C}}(\vec{P}_1) \,|\, \mathcal{F}_{g,n}^{\mathcal{C}}(\vec{P}_2) \rangle = \frac{\delta^{(3g-3+n)}(\vec{P}_1 - \vec{P}_2)}{\rho_{g,n}^{\mathcal{C}}(\vec{P}_1)},\tag{2.21}$$

where the density $\rho_{g,n}^{\mathcal{C}}(\vec{P})$ must be given by the OPE density of the partition function (or correlation function) of the Liouville CFT in the channel $\mathcal{C}$. The channel $\mathcal{C}$ is specified by cutting the (punctured) Riemann surface into $2g - 2 + n$ pairs of pants sewn together along $3g - 3 + n$ internal cuffs and specifying a corresponding dual graph. This is described in detail in section 2.4. In particular, we claim

$$\rho_{g,n}^{\mathcal{C}}(\vec{P}) = \prod_{\text{cuffs } a} \rho_0(P_a) \prod_{\substack{\text{pairs of pants} \\ (i,j,k)}} C_0(P_i, P_j, P_k),\tag{2.22}$$

where $\rho_0$ is the universal Cardy density of states

$$\rho_0(P) = 4\sqrt{2} \sinh(2\pi b P) \sinh(2\pi b^{-1} P),\tag{2.23}$$

and $C_0$ is the universal formula for structure constants defined in eq. (2.17). See also appendix B for more details. We note that the inner product is indeed positive definite, which is the statement that the density $\rho_{g,n}^{\mathcal{C}}(\vec{P})$ is a positive function. This follows immediately from the definition of $C_0$ in eq. (2.17), together with the definition of the double Gamma-function (2.19). Indeed, for real $Q$, one easily checks the properties

$$\overline{\Gamma_b(\tfrac{Q}{2} + iP)} = \Gamma_b(\tfrac{Q}{2} - iP), \qquad\qquad \Gamma_b(Q) > 0,\tag{2.24a}$$

$$\overline{\Gamma_b(Q + iP)} = \Gamma_b(Q - iP), \qquad\qquad \Gamma_b(2Q) > 0.\tag{2.24b}$$

Thus the theory is unitary as long as $c > 1$.

It is sometimes be convenient to work with a basis of conformal blocks that trivializes the inner product of the quantum Teichmüller theory. In particular, one can define a rescaled basis of conformal blocks[10]

$$\left| \widetilde{\mathcal{F}}_{g,n}^{\mathcal{C}}(\vec{P}) \right\rangle = \prod_{\substack{\text{pairs of pants} \\ (i,j,k)}} \sqrt{C_0(P_i, P_j, P_k)} \left| \mathcal{F}_{g,n}^{\mathcal{C}}(\vec{P}) \right\rangle,\tag{2.25}$$

---

[9]Here, we assume that each component of $\vec{P}_1$ and $\vec{P}_2$ is positive. By reflection symmetry, there are also delta-functions of the form $\delta(P_1^a + P_2^a)$ for every component $a$.

[10]Since $C_0(P_1, P_2, P_3) > 0$, the choice of square root is unambiguous.

that is equipped with the inner product

$$\left\langle \widetilde{\mathcal{F}}_{g,n}^{\mathcal{C}}(\vec{P}_1) \,\middle|\, \widetilde{\mathcal{F}}_{g,n}^{\mathcal{C}}(\vec{P}_2) \right\rangle = \frac{\delta^{(3g-3+n)}(\vec{P}_1 - \vec{P}_2)}{\widetilde{\rho}_{g,n}^{\mathcal{C}}(\vec{P}_1)}, \quad \widetilde{\rho}_{g,n}^{\mathcal{C}}(\vec{P}) = \prod_{a=1}^{3g-3+n} \rho_0(P_a). \tag{2.26}$$

We give three different arguments for the validity of eq. (2.22) in this paper. Here we provide a very straightforward argument.

The claim (2.22) follows from the following physical considerations. As we already mentioned, the inner product is the result of a BRST reduction of a "worldsheet theory" consisting of the two conformal blocks and timelike Liouville theory. This is of course not a CFT since it is not crossing symmetric, but otherwise satisfies all other CFT axioms. Performing a BRST reduction ensures that all unphysical states do not propagate. In this context, this hence means that all internal states parametrized by $\vec{P}$ need to be BRST closed. This in particular implies that they need to be level-matched, which shows that the integral is delta-function supported when $\vec{P}_1 = \vec{P}_2$. To determine the density $\rho_{g,n}^{\mathcal{C}}(\vec{P})$, let us first notice that it has to factorize as in (2.22). Indeed, we can think of the target string theory, which is essentially a two-dimensional quantum field theory. The inner product is computing a scattering amplitude in this theory, which can be computed by the Feynman rules. In this context, $\rho_0^{-1}$ plays the role of the propagator, while $C_0^{-1}$ is the structure constant of the trivalent vertices. We have already demonstrated the correctness of $C_0$ by computing the three-point function on the sphere. $\rho_0$ is nothing other than the normalization of the two-point function with these conventions. We have

$$\lim_{P_3 \to \mathbb{1}} C_0(P_1, P_2, P_3) = \rho_0(P_1)^{-1} \delta(P_1 - P_2). \tag{2.27}$$

This leads to the claimed formula. We write here and in the following $P = \mathbb{1}$ as a shortcut for $P = \pm \frac{iQ}{2}$ corresponding to the vacuum $\Delta = 0$. We should notice that it was crucial for this argument that we integrate over Teichmüller space in eq. (2.5) and not the moduli space of surfaces as ordinarily in string theory. Integrating over Teichmüller space leads to a single Feynman diagram in spacetime, while usual string amplitudes would unify all Feynman diagrams of a given loop order into a single diagram (and also avoid UV-divergences).

As promised, we will provide two different derivations below. We can require that the crossing transformations are unitary with respect to this inner product which uniquely fixes it to this form. We will also give an independent argument using 3d gravity.

Let us also comment on the case of the torus without punctures. The inner product is somewhat ill-defined because the partition function of Liouville theory on the torus is ill-defined. However, we can still compute the inner product of two conformal blocks up to an overall ill-defined constant. Conformal blocks on the torus are just Virasoro characters. Writing out (2.5) explicitly for the torus (for which Teichmüller space is the upper half plane) gives

$$\langle \mathcal{F}_{1,0}(P_1) | \mathcal{F}_{1,0}(P_2) \rangle = \int_{\mathbb{H}} \frac{\mathrm{d}^2\tau}{\mathrm{Im}\,\tau} \frac{\bar{q}^{P_1^2} q^{P_2^2}}{|\eta(\tau)|^2} \times |\eta(\tau)|^4 \times \frac{1}{\sqrt{\mathrm{Im}(\tau)}|\eta(\tau)|^2}. \tag{2.28}$$

Here we treated the partition function of timelike Liouville theory just like the partition function of a free boson and avoided the normalization problem. We also set as usual $q = \mathrm{e}^{2\pi i \tau}$. We used the $bc$ ghost partition function $|\eta(\tau)|^4$ on the torus. We also divided by another factor of $\mathrm{Im}\,\tau$. This represents the volume of the residual conformal transformations (conformal Killing vectors). Since Teichmüller space is the coset

$$\frac{\text{metrics on } \Sigma}{\mathrm{Weyl}(\Sigma) \times \mathrm{Diff}_0(\Sigma)}, \tag{2.29}$$

of metrics up to Weyl rescaling and small diffeomorphisms, we should divide by the volume of this residual gauge group just like in string theory. Writing $\tau = x + iy$, we thus obtain with $P_1 \geq 0$, $P_2 \geq 0$,

$$\langle \mathcal{F}_{1,0}(P_1)|\mathcal{F}_{1,0}(P_2)\rangle = \int_{-\infty}^{\infty} \mathrm{d}x \int_0^{\infty} \frac{\mathrm{d}y}{y^{3/2}}\, e^{-2\pi y(P_1^2+P_2^2)+2\pi i x(-P_1^2+P_2^2)} \tag{2.30}$$

$$= \delta(P_1^2 - P_2^2) \int_0^{\infty} \frac{\mathrm{d}y}{y^{3/2}}\, e^{-4\pi y P_1^2} \tag{2.31}$$

$$= \frac{1}{P_1}\delta(P_1 - P_2)P_1 \int_0^{\infty} \frac{\mathrm{d}y}{y^{3/2}}\, e^{-4\pi y}\,. \tag{2.32}$$

The remaining integral strictly speaking does not converge, but we should not assign too much meaning to it, since we anyway already discarded an overall constant. We thus conclude that $\langle \mathcal{F}_{1,0}(P_1)|\mathcal{F}_{1,0}(P_2)\rangle \propto \delta(P_1 - P_2)$.

Let us summarize the discussion of this section. The Hilbert space consisting of all normalizable states is precisely the space of (holomorphic) Liouville conformal blocks,

$$\mathcal{H}_{\mathrm{Vir}} = \{\text{Liouville conformal blocks}\}\,. \tag{2.33}$$

For 3d gravity, we have two copies of this Hilbert space and can think of one set of conformal blocks as left-moving and one as right-moving.

We should also mention that states below and above the threshold have completely different geometric origin. In Teichmüller space, we specify the monodromy of the flat $\mathrm{PSL}(2,\mathbb{R})$ gauge field around the puncture. It can either be an elliptic element of $\mathrm{PSL}(2,\mathbb{R})$, corresponding to a conical defect in the surface, or a hyperbolic element of $\mathrm{PSL}(2,\mathbb{R})$, corresponding to a geodesic boundary of the surface. After quantization, the external conformal weight specifies the type of puncture. A conformal weight below the threshold $\frac{c-1}{24}$ corresponds to a conical defect, while a conformal weight above threshold corresponds to a geodesic boundary. The semiclassical relation between the length of the boundary geodesic and the conformal weight was worked out in [44] (and already conjectured in [29]). It reads

$$\Delta = \frac{c-1}{24} + \left(\frac{\ell}{4\pi b}\right)^2\,, \tag{2.34}$$

i.e. the Liouville momentum $P$ is essentially equal to the geodesic boundary length $\ell$. Some consequences for sub-threshold states and their implications for 3d gravity were discussed in [12, 67].

## 2.4 Mapping class group action and the Moore-Seiberg construction

So far, the quantization procedure and the construction of the Hilbert space is not computationally useful. Indeed, we have assigned an infinite-dimensional Hilbert space to every spatial slice, together with an inner product. However, all infinite-dimensional (separable) Hilbert spaces are isomorphic and thus we haven't gained much at this point. We will in the following again discuss one copy of Teichmüller space. For 3d gravity the whole discussion should be doubled.

To make the construction useful, we have to exhibit more structure on $\mathcal{H}$. The 2d mapping class group $\mathrm{Map}(\Sigma)$ acts on the phase space $\mathcal{T}$. As usual in quantum theory, this means that $\mathcal{H}$ should carry a unitary, projective representation of the mapping class group. Hence for every $\gamma \in \mathrm{Map}(\Sigma)$ there should be a corresponding unitary operator $U(\gamma)$ on $\mathcal{H}$. In fact, we get slightly more than a representation of the mapping class group, we get a representation of the so-called Moore-Seiberg groupoid [45].

To continue, let us recall some facts about the mapping class group $\mathrm{Map}(\Sigma)$. To start with, the mapping class group is generated by the Dehn twists around all the simple closed curves

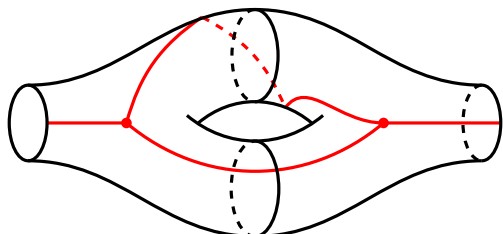

Figure 1: Pair of pants decomposition of a genus 1 surface with two punctures. We also draw the dual graph in red.

on the surface.[11] Let us present the surface in a pair of pants decomposition as in Figure 1. We think of all the punctures as being excised holes in the surface. Then it is in fact the case that the mapping class group is already generated by the Dehn twists around the curves that define the pair of pants decomposition, including the curves around the punctures.[12]

It is also useful to draw a dual graph to the pair of pants decomposition on the surface as indicated in Figure 1. Let us denote the set of all pair of pants decompositions with dual graphs by $X_\Sigma$. It is then clear that the mapping class group acts on the set $X_\Sigma$ in a natural way. This lets one technically turn $X_\Sigma$ into a groupoid, but we will not make use of this terminology.

The group action is not transitive, but there are finitely many orbits. Every orbit can be labelled by a corresponding trivalent graph that is obtained by forgetting the Riemann surface and keeping only the dual graph. For the example of the genus 1 surface with two punctures drawn in Figure 1, there are two such graphs:

$$\text{and} \tag{2.35}$$

It should be clear that this graph remains invariant under the action of the mapping class group.

Every such picture labels precisely one OPE channel for the Virasoro conformal blocks. The construction of Moore and Seiberg establishes a number of basic moves that relate different pictures to each other. Once we know the behaviour of conformal blocks under these basic moves and their consistency conditions, the crossing transformations on arbitrary surfaces can be deduced. We have the following basic moves:

1. Braiding:

$$= \mathbb{B}^3_{12} \tag{2.36}$$

2. Fusion:

$$= \sum_s \mathbb{F}_{st} \begin{bmatrix} 3 & 2 \\ 4 & 1 \end{bmatrix} \tag{2.37}$$

---

[11]In the presence of punctures, we also need rotations around the boundary curves. The relevant group is thus strictly speaking a central extension $\widehat{\text{Map}}(\Sigma)$ of $\text{Map}(\Sigma)$. Mathematically, $\widehat{\text{Map}}(\Sigma)$ is the pure mapping class group (meaning that the mapping group element has to act trivially on the boundary) of the surface, where a small hole around each puncture has been excised. This allows us also consider rotations around the holes. We refer to [45] for more details and suppress the difference of $\text{Map}(\Sigma)$ and $\widehat{\text{Map}}(\Sigma)$ from now on.

[12]One can do slightly better and use even fewer generators [68, 69], but we will not need this.

3. Torus modular S-transform:

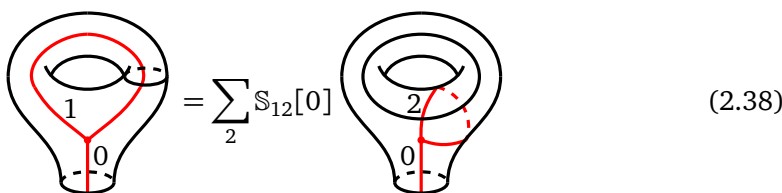

$$\qquad\qquad\qquad (2.38)$$

The basic data $\mathbb{B}$, $\mathbb{F}$ and $\mathbb{S}$ has to satisfy several consistency conditions. To start with, $\mathbb{B}$ is elementary to determine,

$$\mathbb{B}^3_{12} = \pm e^{\pi i(\Delta_3 - \Delta_1 - \Delta_2)}, \qquad\qquad (2.39)$$

which follows from the universal short distance behaviour of the OPE. The sign has to do with (pseudo)reality of the corresponding representations and will be the + sign in our application. Performing the braiding move $\mathbb{B}$ twice as follows leads to a simple Dehn twist:

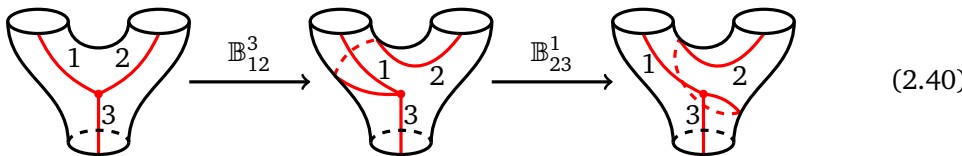

$$\qquad\qquad\qquad (2.40)$$

Thus, it is sometimes useful to also introduce

$$\mathbb{T}_2 = \mathbb{B}^3_{12}\mathbb{B}^1_{23} = e^{-2\pi i \Delta_2}. \qquad\qquad (2.41)$$

The consistency conditions are known as the Moore-Seiberg relations and are listed in appendix A. It was shown in [45, 70] that these relations are complete.

Since 3d gravity is a unitary theory, mapping class group transformations must act unitarily on the Hilbert space. This also follows directly from the abstract definition of the inner product (2.5), since a crossing transformation on the integrand can be undone by changing the integration variable on Teichmüller space.

To summarize, the Hilbert space we get by quantizing Teichmüller space carries a unitary projective representation of the "Moore-Seiberg groupoid"; or, in less fancy language, we can express conformal blocks in one channel through conformal blocks in any other channel, which leads to an action of crossing transformations on the Hilbert space.

## 2.5 Review of the Virasoro crossing kernel

We now discuss the explicit form of the crossing kernels $\mathbb{F}$ and $\mathbb{S}$ that appears in the crossing symmetry of Virasoro conformal blocks.[13]

---

[13]That the crossing kernels satisfy the Moore-Seiberg consistency conditions was established in [51, 71]. We review these consistency conditions in appendix A.

The Virasoro crossing kernels are given by [49, 50]

$$
\mathbb{F}_{P_{21},P_{32}}\begin{bmatrix} P_3 & P_2 \\ P_4 & P_1 \end{bmatrix} = \prod_{\pm_1\pm_2\pm_3=+} \frac{\Gamma_b(\frac{Q}{2}\pm_1 p_2\pm_2 p_3\pm_3 p_{32})\Gamma_b(\frac{Q}{2}\pm_1 p_1\pm_2 p_4\pm_3 p_{32})}{\Gamma_b(\frac{Q}{2}\pm_1 p_1\pm_2 p_2\mp_3 p_{21})\Gamma_b(\frac{Q}{2}\pm_1 p_3\pm_2 p_4\mp_3 p_{21})}
$$

$$
\times \frac{\Gamma_b(Q\pm 2iP_{21})}{\Gamma_b(\pm 2iP_{32})} \int \frac{\mathrm{d}\xi}{i} S_b(\tfrac{Q}{4}+\xi-iP_{1,2,21})S_b(\tfrac{Q}{4}+\xi-iP_{3,4,21})
$$

$$
\times S_b(\tfrac{Q}{4}+\xi-iP_{2,3,32})S_b(\tfrac{Q}{4}+\xi-iP_{1,4,32})S_b(\tfrac{Q}{4}-\xi+iP_{1,21,3,32})
$$

$$
\times S_b(\tfrac{Q}{4}-\xi+iP_{1,2,3,4})S_b(\tfrac{Q}{4}-\xi+iP_{2,21,4,32})S_b(\tfrac{Q}{4}-\xi), \qquad (2.42a)
$$

$$
\mathbb{S}_{P_1,P_2}[P_0] = \frac{e^{\frac{\pi i}{2}\Delta_0}\rho_0(P_2)\Gamma_b(Q\pm 2iP_1)\Gamma_b(\frac{Q}{2}-iP_0\pm 2iP_2)}{S_b(\frac{Q}{2}+iP_0)\Gamma_b(Q\pm 2iP_2)\Gamma_b(\frac{Q}{2}-iP_0\pm 2iP_1)}
$$

$$
\times \int \frac{\mathrm{d}\xi}{i} e^{-4\pi\xi P_1}S_b(\tfrac{Q}{4}+\tfrac{iP_0}{2}\pm iP_2\pm\xi). \qquad (2.42b)
$$

Here, $P_I = \sum_{i\in I} P_i$ for some set of indices $I$. The $\xi$-contour is given by the positively-oriented imaginary axis. Here we have made use of the double sine function $S_b(z)$, related to the double Gamma function as $S_b(z) = \Gamma_b(z)/\Gamma_b(Q-z)$. Due to the reflection formula of the Gamma function, it satisfies the simpler functional equation

$$
S_b(z+b^{\pm 1}) = 2\sin(\pi b^{\pm 1}z)S_b(z), \qquad (2.43)
$$

which together with $S_b(\frac{Q}{2}) = 1$ characterizes it completely. These formulas are obtained by bootstrapping the Moore-Seiberg consistency conditions [45]. As an initial input, one can use the crossing kernels for degenerate Virasoro blocks, which can be expressed explicitly as hypergeometric integrals. The consistency conditions translate then into various shift equations for $\mathbb{F}$ and $\mathbb{S}$, which are solved by (2.42a) and (2.42b).

These formulas hold initially only when all $P_i$'s are real, i.e. $\Delta_i \geq \frac{c-1}{24}$. However, one can extend these formulas meromorphically to any any complex $P_i$. One can in particular also consider the crossing transformation of identity blocks and we have [72]

$$
\mathbb{F}_{P_3,P_{32}}\begin{bmatrix} P_3 & P_2 \\ \mathbb{1} & P_1 \end{bmatrix} = \delta(P_1-P_{32}), \qquad (2.44a)
$$

$$
\mathbb{F}_{\mathbb{1},P_{32}}\begin{bmatrix} P_3 & P_2 \\ P_3 & P_2 \end{bmatrix} = C_0(P_2,P_3,P_{32})\rho_0(P_{32}), \qquad (2.44b)
$$

$$
\mathbb{S}_{\mathbb{1},P}[\mathbb{1}] = \rho_0(P). \qquad (2.44c)
$$

Note that it does not make sense to take outgoing momenta to the identity (nor to any sub-threshold state) since the result of the crossing transformation will always be a linear combination of normalizable blocks. Here $C_0(P_1,P_2,P_3)$ is again the universal DOZZ three-point function in a particularly convenient normalization that we already defined in (2.17). $\rho_0$ is the universal Cardy density of states defined in eq. (2.23). This identity is the main motivation for us to use this convention.

We should also finally mention that it is non-trivial that the crossing matrices indeed preserve the inner product. For example, for the once-punctured torus, this is the statement that

$$
\frac{\delta(P_1-P_2)}{\rho_0(P_1)C_0(P_1,P_1,P_0)} = \langle \mathcal{F}_{1,1}(P_0;P_1)|\mathcal{F}_{1,1}(P_0;P_2)\rangle \qquad (2.45)
$$

$$
\overset{!}{=} \int \mathrm{d}P_1'\,\mathrm{d}P_2'\,\mathbb{S}_{P_1,P_1'}[P_0]\overline{\mathbb{S}_{P_2,P_2'}[P_0]}\langle \mathcal{F}_{1,1}(P_0;P_1')|\mathcal{F}_{1,1}(P_0;P_2')\rangle \qquad (2.46)
$$

$$= \int dP'_1 dP'_2 \, \mathbb{S}_{P_1,P'_1}[P_0] \overline{\mathbb{S}_{P_2,P'_2}[P_0]} \frac{\delta(P'_1 - P'_2)}{\rho_0(P'_1)C_0(P'_1,P'_1,P_0)} \tag{2.47}$$

$$= \int dP \, \frac{\mathbb{S}_{P_1,P}[P_0] \overline{\mathbb{S}_{P_2,P}[P_0]}}{\rho_0(P)C_0(P,P,P_0)}, \tag{2.48}$$

where we renamed $P'_1 \to P$ in the last line. $\mathbb{S}_{P_1,P_2}[P_0]$ behaves in a simple way when exchanging $P_1$ and $P_2$,

$$\frac{\mathbb{S}_{P_1,P}[P_0]}{\rho_0(P)C_0(P,P,P_0)} = \frac{\mathbb{S}_{P,P_1}[P_0]}{\rho_0(P_1)C_0(P_1,P_1,P_0)}. \tag{2.49}$$

This follows from the Moore-Seiberg consistency conditions in appendix A. This allows us rewrite the invariance of the inner produce under crossing symmetry as the condition

$$\int dP \, \mathbb{S}_{P_1,P}[P_0] \overline{\mathbb{S}_{P,P_2}[P_0]} \overset{!}{=} \delta(P_1 - P_2). \tag{2.50}$$

Except for the phase $e^{\frac{\pi i}{2}\Delta_0}$ in (2.42b), $\mathbb{S}$ is real and thus this condition can also be written as

$$\int dP \, \mathbb{S}_{P_1,P}[P_0] \mathbb{S}_{P,P_2}[P_0] \overset{!}{=} e^{\pi i \Delta_0} \delta(P_1 - P_2). \tag{2.51}$$

The condition is one of the Moore-Seiberg consistency conditions (A.5a). It basically says that the square of $\mathbb{S}$ is trivial. A similar argument shows that $\mathbb{F}$ preserves the inner product thanks to the validity of the Moore-Seiberg consistency conditions that we list all in appendix A for convenience. One in particular has to use the fact that the combination

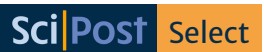 $= \rho_0(P_{32})^{-1} C_0(P_1,P_2,P_{21}) C_0(P_3,P_4,P_{21}) \mathbb{F}_{P_{21},P_{32}} \begin{bmatrix} P_3 & P_2 \\ P_4 & P_1 \end{bmatrix}, \tag{2.52}$

has tetrahedral symmetry as indicated by the picture. This is a consequence of the pentagon equation (A.4). For instance, the fusion kernel is related to its inverse as

$$\frac{\mathbb{F}_{P_{21},P_{32}} \begin{bmatrix} P_3 & P_2 \\ P_4 & P_1 \end{bmatrix}}{\rho_0(P_{32})C_0(P_1,P_4,P_{32})C_0(P_2,P_3,P_{32})} = \frac{\mathbb{F}_{P_{32},P_{21}} \begin{bmatrix} P_3 & P_4 \\ P_2 & P_1 \end{bmatrix}}{\rho_0(P_{21})C_0(P_1,P_2,P_{21})C_0(P_3,P_4,P_{21})}. \tag{2.53}$$

Consistency with unitarity actually fully determines the inner product and can be used to give an independent derivation of eq. (2.21).

## 2.6 The Virasoro TQFT

Let us take stock what has been achieved so far. Quantization of Teichmüller space leads to a Hilbert space consisting of Liouville conformal blocks. This Hilbert space carries a unitary (under the inner product (2.5)) representation of the Moore-Seiberg groupoid, i.e. it closes under crossing transformations.

Let us emphasize that the latter property is very unexpected from a naive 3d point of view. We started with the phase space of $\mathrm{PSL}(2,\mathbb{R})$ Chern-Simons theory and more or less arbitrarily restricted to one component. It is surprising that the quantization obtained from only one

component closes again under crossing. In particular, such a statement can easily be seen to be false for any of the other components of the moduli space of flat $PSL(2, \mathbb{R})$ bundles on the initial value surface $\Sigma$.

This data is enough to define a three-dimensional TQFT (in Euclidean signature). It is of course not a fully conventional TQFT because the Hilbert space is infinite-dimensional. Related to this is also the fact that the trivial Wilson line is not normalizable. This leads to the divergence of some partition functions.

Let us elaborate a little bit more on the construction of this TQFT from the data of Virasoro conformal blocks and their behaviour under crossing transformations. The main goal of any TQFT is to compute partition functions on a 3-manifold $M$, possibly with boundaries and the insertion of Wilson lines. Let us assume that the boundary of $M$ consists of a number of Riemann surfaces, $\partial M = \bigsqcup_{i=1}^{n} \Sigma_i$. The path integral over $M$ then prepares a state in the boundary Hilbert space,

$$Z(M) \in \bigotimes_{i=1}^{n} \mathcal{H}_{\Sigma_i}, \qquad (2.54)$$

i.e. for the Virasoro TQFT, the path integral evaluates to a combination of conformal blocks on the boundary. This can be thought of as the chiral half of the boundary CFT; but of course, being chiral, it is not invariant under modular transformations. When $M$ is closed, the partition function is simply a number.

The standard surgery technique to compute partition functions of TQFTs chops $M$ into smaller and smaller pieces and thus computes $Z(M)$ recursively. If we write $M = M_1 \sqcup M_2$ glued along the common boundary, we have

$$Z(M) = \langle Z(M_1) | Z(M_2) \rangle, \qquad (2.55)$$

where the inner product is taken in the Hilbert space of the joint boundary component. Using the surgery technique, one can derive many seemingly miraculous formulas. However, one has to be careful in the Virasoro TQFT. $Z(M)$ is not guaranteed to give a *normalizable* state in the boundary Hilbert space. Thus the inner product in (2.55) may or may not be finite. Furthermore, we have only defined the Hilbert space for surfaces with negative Euler characteristic and want to avoid cutting along, say, a 2-sphere without punctures.

Doing so gets one immediately into trouble. Indeed, one can for example derive in any TQFT that [46]

$$Z(M_1 \# M_2) = \frac{Z(M_1) Z(M_2)}{Z(S^3)}, \qquad (2.56)$$

where $\#$ denotes the connected sum obtained by cutting out a ball of $M_1$ and $M_2$ and identifying the two manifolds along the 2-sphere boundary created that way. However, the three-sphere partition function that appears is not a well-defined object in the Virasoro TQFT, which we will see in more detail below. In general, we expect that at least partition functions of hyperbolic 3-manifolds make sense in the Virasoro TQFT, since those admit a classical saddle solution in 3d gravity. However, hyperbolic manifolds are always prime, meaning that they cannot be written as a connected sum (except in the trivial way $M \# S^3$). The proof of this statement is very simple and we explain it in appendix C. Thus the left hand side of (2.56) is not a hyperbolic manifold and does not have a well-defined partition function, which is compatible with $S^3$ not having a partition function in the Virasoro TQFT.

When $M$ has a well-defined partition function, we can cut along Riemann surfaces with well-defined Hilbert spaces and reduce the computation of the partition function to a number of elementary building blocks, which can be computed by other means. In this process, we just need a good understanding of the Hilbert space, but no inherently three-dimensional data. In order for the procedure to be consistent, one has to make sure that the result does not depend

on the choices of the cuts in the surgery procedure. This is essentially ensured by the Moore-Seiberg consistency conditions on the 2d mapping class group action. In fact, consistency of the 3d theory often gives simpler ways to demonstrate certain identities for the transformation properties of conformal blocks [52].

Similar to 3d Chern-Simons theory, Virasoro TQFT has a framing anomaly. We only get well-defined results once we thicken the Wilson lines slightly to ribbons, which introduces a self-linking number. Changing the self-linking number by one unit multiplies the partition function by $e^{2\pi i \Delta}$, where $\Delta$ is the conformal weight associated to the Wilson line. Similarly, one has to specify a framing of the full three-manifold and change of framing multiplies the partition function by $e^{\frac{\pi i c}{12}}$. However, the framing anomaly will cancel in 3d gravity, where we need two copies of Virasoro TQFT. The second copy has a reversed orientation which means that all phases are opposite. Notice that we can have in principle spinning Wilson lines carrying a different left- and right-moving conformal weight. Cancellation of the framing anomaly requires that

$$\Delta - \widetilde{\Delta} \in \mathbb{Z}. \tag{2.57}$$

The relation to 3d gravity makes the semiclassical behaviour of the partition functions particularly transparent. As was already mentioned, $b^2$ plays to role of $\hbar$ in the quantization of Teichmüller space. Thus, we expect that for a closed hyperbolic manifold

$$|Z(M)| = e^{-\frac{1}{2\pi b^2}\text{vol}(M) + \mathcal{O}(1)}, \tag{2.58}$$

where $\text{vol}(M)$ is the hyperbolic volume of $M$.[14] The factor $2\pi$ is conventional and related to the normalization of the hyperbolic metric. This property is known as the volume conjecture [73–75]. There can be a large phase in this expression which cancels out once one combines left- and right-movers. Virasoro TQFT and its relation to gravity actually predicts a generalization of this conjecture for the first subleading correction and the case of boundaries. We discuss it further in the discussion section 4.

We should also mention that a similar TQFT based on the quantization of Teichmüller space was defined by Andersen and Kashaev [31, 32]. Their theory is defined in an ad-hoc way by introducing certain special classes of triangulations. They also coined the name Teichmüller TQFT for the theory. Our discussion is in our view physically much better motivated and should be equivalent to their theory, but this is not completely obvious. The transformation between the two different quantizations was discussed also in [30]. As mentioned in the Introduction, we thus prefer to call the theory in this paper Virasoro TQFT.

## 2.7 A relation between 3d gravity and Virasoro TQFT

After our somewhat long detour into the Virasoro TQFT, we now want to formulate a precise relation between 3d gravity and two copies of Virasoro TQFT that forms the heart of the paper. We should note that while we have discussed 3d gravity in Lorentzian signature so far, we will now change to Euclidean quantum gravity. It was necessary to start in Lorentzian signature to determine the correct phase space and Hilbert space. The Hilbert space is however associated to the initial value surface and should thus not depend on the spacetime signature. We can thus take it as a starting point for the construction of Euclidean quantum gravity partition functions.

In Euclidean 3d gravity, we would like to compute the path integral

$$\sum_{\text{topologies}} \int \frac{[dg]}{\text{Diff}(M, \partial M)} e^{-S[g]}, \tag{2.59}$$

---

[14]Three-dimensional hyperbolic manifolds are rigid, in the sense that they only admit one hyperbolic structure. Thus the hyperbolic volume becomes a topological invariant of $M$.

over all metrics with given boundary conditions. We recall from footnote 3 that we only consider orientable manifolds in this sum and only gauge orientation-preserving diffeomorphisms. $\text{Diff}(M, \partial M)$ is the group of all diffeomorphisms – large and small – that map each boundary component to itself (but that don't necessarily act trivially on it). This still differs from what is done in Virasoro TQFT, where we only mod out by the small component of $\text{Diff}(M, \partial M)$.[15] The relative mapping class group is now defined as

$$\text{Map}(M, \partial M) = \text{Diff}(M, \partial M)/\text{Diff}_0(M, \partial M),\qquad(2.60)$$

and represents the mismatch of the gravity gauge group and the TQFT gauge group. Similarly, there are large diffeomorphisms that act as modular transformations on the boundary and hence do not change the boundary conditions, but potentially change the bulk manifold. They are part of the sum over topologies in the gravitational path integral. These large diffeomorphisms are themselves parametrized by elements in the boundary mapping class group. Thus the gravity partition function includes an additional sum over the boundary modular transformations $\gamma \in \text{Map}(\partial M)$.[16]

For a fixed topology $M$, this gives the formal expression

$$Z_{\text{grav}}(M) = \frac{1}{|\text{Map}(M, \partial M)|} \sum_{\gamma \in \text{Map}(\partial M)} |Z_{\text{Vir}}(M^\gamma)|^2 \,.\qquad(2.61)$$

This is not yet in its final form since $|\text{Map}(M, \partial M)|$ can be infinite. However, $\text{Map}(M, \partial M)$ naturally maps to $\text{Map}(\partial M)$ since any mapping class group transformation of the 3d manifold gives also a mapping class group transformation of the boundary. For $\gamma \in \text{Map}(M, \partial M) \subset \text{Map}(\partial M)$, we have

$$Z_{\text{Vir}}(M^\gamma) = Z_{\text{Vir}}(M)\,.\qquad(2.62)$$

Thus we arrive at the final formula

$$Z_{\text{grav}}(M) = \frac{1}{|\text{Map}_0(M, \partial M)|} \sum_{\gamma \in \text{Map}(\partial M)/\text{Map}(M, \partial M)} |Z_{\text{Vir}}(M^\gamma)|^2 \,.\qquad(2.63)$$

Here,

$$\text{Map}_0(M, \partial M) = \frac{\text{Map}(M, \partial M)}{\ker(\text{Map}(M, \partial M) \longrightarrow \text{Map}(\partial M))}\,,\qquad(2.64)$$

is the the part of the bulk mapping class group that acts trivially on the boundary.

As a trivial example, let us decode the sum in eq. (2.63) for thermal AdS$_3$, which is topologically a solid torus. We have $\text{Map}(\partial M) = \text{PSL}(2, \mathbb{Z})$ – the standard mapping class group of the torus. The bulk mapping class group is generated by Dehn twists along the contractible bulk disk (a so-called compressible disk) and is isomorphic to $\mathbb{Z} \subset \text{PSL}(2, \mathbb{Z})$, see Figure 2. Thus eq. (2.63) just turns into the standard Maloney-Witten sum [9].

In general, this formula is not well-defined since both the ingredients $Z_{\text{Vir}}(M)$ and $|\text{Map}_0(M, \partial M)|$ may diverge. A simple example where this happens is for $M = \Sigma \times S^1$, where $\Sigma$ is a Riemann surface, possibly with boundaries. Then, assuming that $\Sigma$ is not a torus, the 3-dimensional mapping class group of $M$ is just the two-dimensional mapping class group of

---

[15]In the 2d context, this is often called the pure mapping class group, since we do not allow for different boundary components to be permuted under the mapping class group action.

[16]In the case where where punctures are present in the boundary, this is strictly speaking the so-called pure mapping class group that does not allow for punctures to be permuted. We should also note that contrary to the discussion in section 2.4, $\text{Map}(\partial M)$ denotes the actual mapping class group and not a central extension thereof. This reflects the fact that the framing anomaly between left- and right-movers cancels.

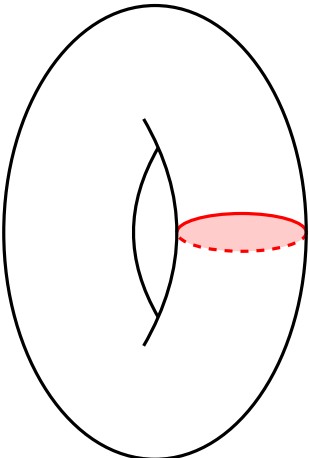

Figure 2: The mapping class group of thermal AdS$_3$ is generated by Dehn twists around the contractible disk drawn in red. This means that we cut the solid torus along the red disk, twist it by 360 degrees and then glue it back.

$\Sigma$, which is usually infinite. We also have $Z_{\text{Vir}}(\Sigma \times \text{S}^1) = \dim \mathcal{H}_\Sigma = \infty$ and thus (2.63) is ill-defined.[17]

However, we now explain that (2.63) is useful for "most" three-manifolds. In particular, it is well-defined for all hyperbolic three-manifolds, i.e. all manifolds admitting a classical gravity solution. Similarly to the two-dimensional situation, it is a well-known folklore that most three-manifolds are hyperbolic, though it is difficult to quantify this. See e.g. [76–79] for evidence in this direction. Indeed, it is a non-trivial fact that the mapping class group $\text{Map}(M)$ is finite for all closed hyperbolic 3-manifolds, while $\text{Map}_0(M, \partial M)$ is trivial for all hyperbolic 3-manifolds with boundary. This is in very stark contrast to the two-dimensional situation. This is a corollary of the rigidity theorem for three-dimensional hyperbolic manifolds. The basic Mostow rigidity theorem says that a closed hyperbolic manifold admits exactly one hyperbolic structure, which is hence rigid. The theorem was generalized by Bers, Maskit and Sullivan [80] to the case with boundaries. Assuming a suitable finiteness condition on the manifold, the theorem states that the bulk hyperbolic metric is fully determined by the boundary hyperbolic metrics (and a choice of boundary mapping class group element). We give more details in appendix C.

Let us explain why $|\text{Map}(M)| < \infty$ for a closed hyperbolic three-manifold. We explain the case with boundaries in appendix C, in which case we actually have $|\text{Map}_0(M, \partial M)| = 1$. Let $[\gamma] \in \text{Map}(M)$ be a mapping class group element. Because of Mostow rigidity, we can choose $\gamma$ to be an isometry, which gives a canonical representative. Thus, for closed hyperbolic three-manifolds,[18]

$$\text{Map}(M) \subset \text{Isom}(M). \tag{2.65}$$

However, $\text{Isom}(M)$ is a discrete group and the discrete part of the isometry group of a compact manifold is necessarily finite.[19] The argument for manifolds with boundaries is similar, but slightly more involved. Thus, to summarize, we can simplify eq. (2.63) in the hyperbolic case to the equations given in the introduction, (1.1), while the case without boundaries simply reads

---

[17]In this case, it is actually understood that $Z_{\text{grav}}(\Sigma \times \text{S}^1)$ still diverges and the factor of $|\text{Map}_0(M, \partial M)|^{-1}$ on the right hand side of (2.63) is only enough to remove the divergence of one chiral half of the partition function [54].

[18]It is actually an isomorphism.

[19]In general, the isometry group of a Riemannian manifold is a Lie group and is compact when $M$ is compact [81]. Since the isometry group is discrete in our case, it follows that it is finite.

$$Z_{\text{grav}}(M) = \frac{1}{|\text{Map}(M)|} |Z_{\text{Vir}}(M)|^2.\qquad(2.66)$$

The other important component entering (1.1) and (2.66) is the Virasoro TQFT partition function. As already mentioned, we expect it to be well-defined if $M$ is hyperbolic, since we could evaluate the gravitational path integral semiclassically in this case. In particular, the volume conjecture (2.58) gives the semi-classical behaviour of the partition function.

Finally, we can of course ask whether the sum over $\text{Map}(\partial M)/\text{Map}(M, \partial M)$ in (2.63) converges. In simple examples, such as the solid torus, the answer is known to be negative, but the sum can be regulated [9,20]. We leave it for future investigation to study this sum in more general cases.

We should also remark on the appearance of holographic renormalization and the conformal anomaly in this framework. When computing partition functions on manifolds $M$ with an AdS boundary in this framework, we obtain a combination of conformal blocks on the boundary $\partial M$, as expected by the AdS/CFT correspondence. In the metric formalism of gravity, one however has to be careful how to even define the tree-level action on a non-compact manifold. This usually proceeds by choosing some cut-off surface far out near the boundary of $M$ and cancelling divergent terms by suitable counterterms on the cut-off surface. The value of the on-shell action depends on the choice of the cut-off surface, which leads to the appearance of the conformal anomaly. This process is known as holographic renormalization [82] and can be carried out very explicitly in $\text{AdS}_3$ [83]. In our framework, it is not necessary to use such an adhoc procedure of choosing a cutoff surface – the partition functions are well-defined without any regularization. However, the fact that conformal blocks transform non-trivially under a change of metric reflects of course the same conformal anomaly that is present in the usual metric formalism.

To summarize, (1.1) gives a simple way to compute the gravity partition function on a fixed hyperbolic 3-manifold. We turn it into a concrete algorithm below and compute various examples in our follow-up paper [52]. It trivializes many of the computations that have been done in the literature.

## 2.8 Two-dimensional analogue

Let us make contact with better-known statements and explain a precise analogue for 2d gravity. We consider JT gravity, a particular dilaton-gravity theory, see e.g. [4] for a review. This theory is also classically equivalent to a gauge theory – $\text{PSL}(2,\mathbb{R})$ $BF$ theory [84] with action

$$S = \frac{1}{4\pi G_{\text{N}}} \int_{\Sigma} \text{tr}(BF),\qquad(2.67)$$

where $B$ is an adjoint scalar and $F = \mathrm{d}A + A \wedge A$ the curvature of the $\text{PSL}(2,\mathbb{R})$ gauge field. JT gravity can be obtained from 3d gravity by dimensional reduction and this relation is the shadow of the relation between 3d gravity and $\text{PSL}(2,\mathbb{R})$ Chern-Simons theory.

This relation has the same problems as discussed in the three-dimensional setting above. In particular, most $\text{PSL}(2,\mathbb{R})$ gauge fields would look very singular from a gravity point of view. Different approaches to partially cure this were discussed in the literature [85,86].

Computing partition functions of $BF$-theory is very simple. $B$ acts as a Lagrange multiplier and can be integrated out, which imposes $F = 0$. As a consequence, the path integral computes the volume of the moduli space $\mathcal{M}_{\text{PSL}(2,\mathbb{R})}^{\text{flat}}$ of flat $\text{PSL}(2,\mathbb{R})$-bundles on a Riemann surface,[20]

$$Z_{\Sigma} = \text{vol}(\mathcal{M}_{\Sigma}).\qquad(2.68)$$

---

[20]This requires a careful treatment of the determinants of the Fadeev-Popov ghosts [2]. In particular, this analysis also determines the measure on moduli space to be the Weil-Petersson volume form.

We have been purposefully somewhat vague about the precise definition of the moduli space $\mathcal{M}_\Sigma$.

As we have mentioned in section 2.2, the moduli space of flat $\mathrm{PSL}(2, \mathbb{R})$ bundles is disconnected and the component corresponding to smooth metrics is the Teichmüller component. Thus the two-dimensional analogue of passing from $\mathrm{PSL}(2, \mathbb{R})$ Chern-Simons theory to Virasoro TQFT is to replace the moduli space of flat $\mathrm{PSL}(2, \mathbb{R})$ bundles by Teichmüller space.

We should mention that one can see this also from the perspective of canonical quantization of JT gravity – just like we did above for 3d gravity. Indeed, we would like to impose that the metric on every one-dimensional spatial slice in JT gravity is smooth which is tantamount to saying that all $\mathrm{PSL}(2, \mathbb{R})$ holonomies around the cycles of the surface should be hyperbolic. This uniquely picks out the Teichmüller component.

Passing to "Teichmüller $BF$ theory" still treats the mapping class group incorrectly. Indeed, before dividing by the mapping class group, all partition functions are divergent since Teichmüller space is non-compact and thus $\mathrm{vol}(\mathcal{T}) = \infty$. The correct gravity partition function is given by

$$\frac{1}{|\mathrm{Map}(\Sigma)|} \mathrm{vol}(\mathcal{T}) \simeq \mathrm{vol}(\mathcal{T}/\mathrm{Map}(\Sigma)). \tag{2.69}$$

Of course, the left-hand side of this equation does not make sense since $\mathrm{Map}(\Sigma)$ is typically an infinite group. However, $\mathcal{T}/\mathrm{Map}(\Sigma)$ is the moduli space of Riemann surfaces which famously has a finite volume under the Weil-Petersson measure [55].

Thus, while the logic of the interplay of gravity and gauge theory in two dimensions is essentially identical, the technical details especially in the last step are quite different because $\mathrm{Map}(\Sigma)$ is not a finite group.

## 3 Partition functions on hyperbolic 3-manifolds

We will now explain how to use the structure explained in the previous section to compute the gravity partition function for simple topologies. We will also explain a completely algorithmic way to compute the partition function of any hyperbolic three-manifold in this manner.

### 3.1 The inner product of Virasoro TQFT and the Euclidean wormhole

In section 2.3 we gave an argument for the explicit form of the inner product (2.21) and (2.22) in the Hilbert space of the quantum Teichmüller theory by appealing to the string worldsheet interpretation of the conformal block inner product as put forward by H. Verlinde in [29]. Here we will describe a bootstrap argument for this inner product that connects the inner product with the path integral of AdS$_3$ Einstein gravity (possibly coupled to massive point particles) on the Euclidean wormhole with the topology $\Sigma_{g,n} \times I$, where $\Sigma_{g,n}$ is a (possibly punctured) Riemann surface and $I = [0, 1]$ is the unit interval.

Consider Virasoro TQFT on the Euclidean wormhole $\Sigma_{g,n} \times I$. If $\Sigma_{g,n}$ is a punctured Riemann surface, then there are Wilson lines extending into the bulk of the wormhole. Because Virasoro TQFT is a topological theory, we can freely shrink the length of the interval to zero, in which case the partition function should compute a modular invariant partition function on $\Sigma_{g,n}$. This situation is familiar from the relationship between Chern-Simons theory and WZW models based on a compact group, where the path integral of the Chern-Simons theory of $\Sigma_{g,n} \times I$ implements the sum over paired (holomorphic times anti-holomorphic) conformal blocks that computes the partition function of the WZW model on $\Sigma_{g,n}$ [87].

In our case, the path integral of Virasoro TQFT on the Euclidean wormhole implements the resolution of the identity in the Hilbert space of conformal blocks. Indeed, given a three-manifold with appropriate boundaries and Wilson line configurations, one may always cut

along $\Sigma_{g,n}$ in the bulk and glue in the Euclidean wormhole without changing the partition function.

The Euclidean wormhole prepares a state in two copies of the Hilbert space associated with the $\Sigma_{g,n}$ boundaries $\mathcal{H}_{\Sigma_{g,n}} \otimes \mathcal{H}_{\Sigma_{g,n}}$. Hence the TQFT partition function must be written as a linear combination of products of Virasoro conformal blocks on $\Sigma_{g,n}$. When we resolve the identity on the Hilbert space $\mathcal{H}_{\Sigma_{g,n}}$ associated with one of the boundaries, the structure of the inner product (2.21) implies that we get a pairing between conformal blocks that is diagonal in the internal conformal weights: in other words, the partition function on the Euclidean wormhole must take the form

$$Z_{\text{Vir}}(\Sigma_{g,n} \times I) = \int d^{3g-3+n}\vec{P} \; \rho_{g,n}^{\mathcal{C}}(\vec{P}) \left| \mathcal{F}_{g,n}^{\mathcal{C}}(\vec{P}) \right\rangle \otimes \left| \mathcal{F}_{g,n}^{\mathcal{C}}(\vec{P}) \right\rangle. \tag{3.1}$$

This form of the wormhole partition function is enough to argue for (2.22). The argument is simple. The partition function of the TQFT on the Euclidean wormhole must compute a unitary partition function invariant under the action of the mapping class group on $\Sigma_{g,n}$ (in other words, we must get the same answer no matter which channel $\mathcal{C}$ we choose in the conformal block decomposition (3.1)). In fact, there is significant evidence that Liouville CFT is the unique[21] unitary solution to the crossing equations with $c > 1$ and only scalar Virasoro primary operators in its spectrum [28, 88, 89]. So we have

$$Z_{\text{Vir}}(\Sigma_{g,n} \times I) = \left| Z_{\text{Liouville}}(\Sigma_{g,n}) \right\rangle, \tag{3.2}$$

i.e. $\rho_{g,n}^{\mathcal{C}}(\vec{P})$ is the OPE density of Liouville CFT on $\Sigma_{g,n}$. This result also holds in the presence of boundary operator insertions with conformal weights below the $\frac{c-1}{24}$ threshold (corresponding to Wilson lines in the TQFT with elliptic rather than hyperbolic holonomy), in which case the right-hand side of (3.2) is the analytic continuation of the observable in the Liouville CFT.[22]

So far we have been intentionally vague about the moduli of the $\Sigma_{g,n}$ boundaries. Indeed, it is not essential for the purposes of this discussion that the moduli on the two boundaries are identical; the same bootstrap argument fixes $\rho_{g,n}^{\mathcal{C}}(\vec{P})$ in terms of the OPE density of the Liouville CFT regardless of the kinematic configuration. Collectively denoting the moduli of the $\Sigma_{g,n}$ boundaries by $\mathbf{m}_i$, we have[23]

$$Z_{\text{Vir}}(\Sigma_{g,n} \times I; \mathbf{m}_1, \mathbf{m}_2) = \left\langle \mathbf{m}_1, \mathbf{m}_2 \left| Z_{\text{Liouville}}(\Sigma_{g,n}) \right\rangle = Z_{\text{Liouville}}(\Sigma_{g,n}; \mathbf{m}_1, \mathbf{m}_2). \tag{3.4}$$

So in the case of different moduli on the two ends of the wormhole, the Virasoro TQFT partition function is given by the analytic continuation of the Liouville CFT partition function on $\Sigma_{g,n}$ away from Euclidean kinematics. We should also mention that the Liouville partition function is usually thought of as being holomorphic in $\mathbf{m}_1$, but anti-holomorphic in $\mathbf{m}_2$. One may equally well take the conformal blocks associated to both boundaries as being holomorphic. The difference lies in the orientation of the boundary surfaces: inward-pointing boundary orientations lead to holomorphic blocks, while outward-pointing boundary orientations lead to antiholomorphic blocks.

---

[21]Technically the statement is that observables in Liouville theory are the unique solutions to the scalar-only crossing equations up to normalization of the external operators and up to the possibility of tensoring with a 2d TQFT.

[22]This analytic continuation may require the deformation of the contour integral over internal Liouville momenta $\vec{P}$ in (3.1).

[23]Here we are projecting the wormhole partition function onto a state $\langle \mathbf{m}_1, \mathbf{m}_2 | = \langle \mathbf{m}_1 | \otimes \langle \mathbf{m}_2 |$ with definite moduli on the two boundaries. In particular we have

$$\left\langle \mathbf{m} \left| \mathcal{F}_{g,n}^{\mathcal{C}}(\vec{P}) \right\rangle = \mathcal{F}_{g,n}^{\mathcal{C}}(\vec{P}; \mathbf{m}). \tag{3.3}$$

The result (3.4) relating the partition function of Virasoro TQFT on the Euclidean wormhole to the Liouville partition function could have been anticipated from recent results on wormholes in 3d gravity. Indeed, the path integral of semiclassical $AdS_3$ gravity (possibly coupled to massive particles) on the Euclidean wormhole $\Sigma_{g,n} \times I$ was recently computed in [15], where it was shown that the wormhole partition function is given by the square of the corresponding observable on $\Sigma_{g,n}$ in Liouville CFT with the moduli of the two boundaries paired as in (3.4), which in turn matched with the average of a product of CFT obvservables on $\Sigma_{g,n}$ with a Gaussian random ansatz for the structure constants.

It is now straightforward to write out the prescription (1.1) to obtain the gravity partition function in this case. The boundary mapping class group is given by $\mathrm{Map}(\Sigma_{g,n}) \times \mathrm{Map}(\Sigma_{g,n})$, while the bulk mapping class group is given by the diagonal subgroup. Thus the modular sum runs over

$$\gamma \in \big(\mathrm{Map}(\Sigma_{g,n}) \times \mathrm{Map}(\Sigma_{g,n})\big)/\mathrm{Map}(\Sigma_{g,n})\,. \tag{3.5}$$

We can choose the representative in the coset that only acts non-trivially on the right boundary. Hence

$$Z_{\mathrm{grav}}(\Sigma_{g,n} \times I) = \sum_{\gamma \in \mathrm{Map}(\Sigma_{g,n})} |Z_{\mathrm{Liouville}}(\Sigma_{g,n}; \mathbf{m}_1, \gamma \cdot \mathbf{m}_2)|^2\,. \tag{3.6}$$

As mentioned around eq. (2.57) the Wilson lines connecting the two boundaries can be spinning in which case the conformal weights of the two Liouville correlation functions entering (3.6) are different.

This is of a similar form as the result obtained by Cotler and Jensen for the torus wormhole $\mathbb{T}^2 \times I$ [13]. However, the similarity is only superficial. Our formalism predicts that the torus wormhole is divergent due to the continuum of states in the Liouville theory. Dividing out this divergent prefactor, the Liouville torus partition function just becomes that of a free noncompact boson

$$Z_{\mathrm{boson}}(\tau) = \frac{1}{\sqrt{2\,\mathrm{Im}\,\tau}\,|\eta(\tau)|^2} = \frac{1}{\sqrt{-i(\tau - \bar{\tau})}\,\eta(\tau)\eta(-\bar{\tau})}\,. \tag{3.7}$$

We then go away from Euclidean kinematics and replace $\tau \to \tau_1$, $\tau \to -\bar{\tau}_2$, corresponding to the orientation reversal as discussed above. Taking the absolute value squared and summing over modular images of $\tau_2$ gives

$$Z_{\mathrm{grav}}(\mathbb{T}^2 \times I) \propto \frac{1}{2} \sum_{\gamma \in \mathrm{PSL}(2,\mathbb{Z})} \frac{1}{|\tau_1 + \gamma \cdot \tau_2|\,|\eta(\tau_1)|^2\,|\eta(\gamma \cdot \tau_2)|^2} \tag{3.8}$$

$$= Z_{\mathrm{boson}}(\tau_1) Z_{\mathrm{boson}}(\tau_2) \sum_{\gamma \in \mathrm{PSL}(2,\mathbb{Z})} \frac{\sqrt{\mathrm{Im}(\tau_1)\,\mathrm{Im}(\tau_2)}}{|\tau_1 + \gamma \cdot \tau_2|}\,. \tag{3.9}$$

This is similar, but not identical, to the result of [13]. The difference can be traced to a different measure in the integral over the intermediate $P$'s in (3.1). While the measure is flat in $P$ in our formalism for the case of the torus, see eq. (2.32), it is flat in the conformal weight $\Delta$ in the formalism of [13], accompanied by an additional overall factor of $\sqrt{\mathrm{Im}\,\tau_1\,\mathrm{Im}\,\tau_2}$ so that the result is invariant under diagonal modular transformations. Of course, the torus wormhole is not a hyperbolic manifold — it is not a classical solution of Einstein's equations — so perhaps it is not surprising that there are ambiguities in the computation of the path integral. It would be good to understand this mismatch better.

## 3.2 Handlebodies

Another very basic class of 3-manifold is provided by handlebodies (genus-$g$ Riemann surfaces $\Sigma_g$ with some cycles filled in). Thus let us discuss the Virasoro TQFT and gravity partition

function on handlebodies. The filled-in cycles determine a conformal block channel $\mathcal{C}$ in the boundary Riemann surface and thus we will denote a handlebody by $\mathsf{S}\Sigma_g^{\mathcal{C}}$.[24]

For an empty handlebody, the TQFT path integral on a handlebody $\mathsf{S}\Sigma_g^{\mathcal{C}}$ is given by the Virasoro identity block in the channel $\mathcal{C}$:

$$Z_{\text{Vir}}(\mathsf{S}\Sigma_g) = \left| \mathcal{F}_g^{\mathcal{C}}(\mathbb{1}) \right\rangle. \tag{3.10}$$

This follows from the quantization of the appropriate coadjoint orbit of the Virasoro group, see e.g. [10,90]. More generally, we can consider the TQFT path integral on a handlebody with a network of Wilson lines threading cuffs in the pair of pants decomposition of the (possibly punctured) handlebody. In this case, the TQFT partition function computes the corresponding Virasoro conformal block in the channel $\mathcal{C}$ specified by the Wilson line network

$$Z_{\text{Vir}}\left( \vcenter{\hbox{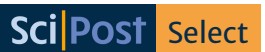}} \right) = \left| \mathcal{F}_{g,n}^{\mathcal{C}}(\vec{P}) \right\rangle. \tag{3.11}$$

Let us recall that Wilson lines are characterized semiclassically by the monodromies of the flat $\text{PSL}(2,\mathbb{R})$ gauge field around the lines, which in turn are labelled by maps $\rho$ from $\pi_1(\Sigma)$ into $\text{PSL}(2,\mathbb{R})$ up to conjugation. These are labelled by the Liouville momenta $P$ as in (2.34).

That the path integral on handlebodies with Wilson lines in the bulk computes a conformal block associated with the boundary surface is a familiar fact from the Chern Simons/WZW model correspondence [87]. Up to overall normalization, this follows again for example from the quatization of Virasoro coadjoint orbits of [90]. The disk punctured by a Wilson line with momentum $P$ gives rise to the corresponding Virasoro representation on the boundary [87]. This tells us that the momentum flowing through the dotted disks in (3.11) is fixed and hence the corresponding states has to be proportional to the conformal block in the boundary. The normalization can be determined by pinching all these disks – this degenerates the handlebody to three-punctured spheres that only touch at the insertion points of the Wilson lines. This then reduces the computation to determining the normalization for a three-punctured sphere

$$\tag{3.12}$$

which gives a state in the three-punctured sphere Hilbert space. This is a one-dimensional Hilbert space, so only the normalization is meaningful. It is convenient to define the trivalent junction such that the corresponding state has unit normalization as in footnote 7. This endows the conformal blocks with a natural normalization that is commonly adopted in the CFT literature. Let us explain this a bit more in detail. We can define a trivalent junction equivalently by carving out a spherical boundary around the junction and dividing the result by the $C_0(P_1, P_2, P_3)$:

$$\tag{3.13}$$

24$\mathcal{C}$ is not uniquely specified from the bulk topology, since we can for example perform Dehn twists along the filled-in cycles without changing the topology. This is of course precisely the role of the bulk mapping class group that we will discuss below.

Indeed, this follows immediately by comparison with our earlier computation of the Euclidean wormhole which in turn followed directly from the normalization of the inner product (2.21). Comparing the unit normalization of (3.12) with (3.4)

$$Z_{\mathrm{Vir}}\left(\begin{array}{c}P_1 \quad P_2 \\ P_3\end{array}\right) = Z_{\mathrm{Liouville}}(\Sigma_{0,3}) = C_0(P_1, P_2, P_3), \qquad (3.14)$$

gives the relation (3.13).

The identity blocks and indeed all conformal blocks with any internal conformal weights below the $\frac{c-1}{24}$ threshold are non-normalizable states in the Hilbert space of Virasoro TQFT equipped with the inner product (2.5).[25] Thus the TQFT path integral is not guaranteed to prepare normalizable states in the corresponding boundary Hilbert spaces. We will have more to say about this in section 3.5.

One can straightforwardly apply the prescription (1.1) to turn this computation into a full-fledged gravity partition function. Let us focus on the empty handlebody. $\mathrm{Map}(S\Sigma_g, \Sigma_g)$ is known as the handlebody group $\mathcal{H}_g$. It is a relatively complicated group; in particular, while being finitely generated, it is not only generated by Dehn twists around curves bounding disks in the bulk [91]. It is also a fact that $\mathcal{H}_g \subsetneq \mathrm{Tor}(\Sigma_g)$ [92, Theorem 3.2]. $\mathrm{Tor}(\Sigma_g)$ is the Torelli subgroup of the mapping class group given by the kernel of the natural map $\mathrm{Map}(\Sigma_g) \longrightarrow \mathrm{Sp}(2g, \mathbb{Z})$ obtained by reducing to homology. The image of $\mathcal{H}_g$ in $\mathrm{Sp}(2g, \mathbb{Z})$ is given by the subgroup

$$\Gamma_\infty = \left\{ \begin{pmatrix} A & B \\ 0 & D \end{pmatrix} \,\middle|\, AD^{\mathsf{T}} = \mathbb{1}, \; B^{\mathsf{T}} D = D^{\mathsf{T}} B \right\}. \qquad (3.15)$$

As a consequence $\mathrm{Map}(\Sigma_g)/\mathcal{H}_g$ is strictly bigger than the coset $\mathrm{Sp}(2g, \mathbb{Z})/\Gamma_\infty$ and the modular sum is bigger. This differs slightly from what was previously proposed in the literature [8]. Summarizing, we have

$$Z_{\mathrm{grav}}(S\Sigma_g) = \sum_{\gamma \in \mathrm{Map}(\Sigma_g)/\mathcal{H}_g} \left| \mathcal{F}_g^{\gamma \cdot \mathcal{C}}(\vec{\mathbb{1}}) \right|^2. \qquad (3.16)$$

## 3.3  Naive surgery

To compute partition functions of Virasoro TQFT on arbitrary (hyperbolic) three-manifolds $\mathcal{M}$, we seek a procedure that allows us to cut $\mathcal{M}$ into a number of smaller elementary building blocks on which we understand how to compute the TQFT partition function, which we later assemble together according to the rules of the TQFT. For TQFTs based on modular tensor categories (MTCs), this is a well-developed subject, and a variety of techniques are available for the computation of such partition functions. But the Virasoro TQFT is not a conventional TQFT: it admits a continuous spectrum of Wilson lines, and the trivial line is not in the spectrum of the theory. Here we will see that many of the techniques familiar from more conventional 3d TQFTs will give nonsensical results when naively applied to the Virasoro TQFT for reasons related to these peculiar aspects of the TQFT. Nonetheless, in what follows we will argue that these pathologies can be overcome and that standard techniques are available that facilitate the unambiguous determination of the partition functions of Virasoro TQFT on arbitrary hyperbolic three-manifolds.

---

[25]Of course, as demonstrated in (2.21), even the conformal blocks with all internal weights above the $\frac{c-1}{24}$ threshold are only delta-function normalizable states in the boundary Hilbert space due to the continuous spectrum.

Partition functions of Virasoro TQFT on three-manifolds should be regarded as states in the Hilbert space associated with the asymptotic boundaries as in (2.54). When we cut the manifold $M$ into smaller pieces $M_1$ and $M_2$, we introduce a new boundary and the partition function $Z(M)$ is computed by the inner product of the partition functions $Z(M_1)$ and $Z(M_2)$ in the Hilbert space associated with this boundary as in eq. (2.55). Thus when decomposing $M$, we must ensure that we are only cutting along surfaces on which the Hilbert space of Virasoro TQFT is well-defined. A priori, this requires that we only cut along surfaces $\Sigma_{g,n}$ with negative Euler characteristic

$$2 - 2g - n < 0 \, . \tag{3.17}$$

In particular the Hilbert spaces associated with the sphere with 0, 1, or 2 punctures are not well-defined in Virasoro TQFT, and so in computing partition functions we should not cut along these surfaces. In these cases the Liouville OPE density that appears in the inner product (2.21) is not well-defined. For the sphere with zero punctures, this amounts to the familiar statement that the identity is not a normalizable operator (not even delta-function normalizable) in the Hilbert space of Liouville CFT. Similarly, the sphere one-point function of local operators vanishes due to conformal symmetry, and the sphere two-point function is only delta-function normalizable.

Let us briefly pause to comment on the case of the torus with zero punctures, which obviously does not satisfy (3.17). In this case the Hilbert space is actually well-defined as a vector space — in particular, it is identified with the space of Virasoro torus characters with conformal weight above the $\frac{c-1}{24}$ threshold — but there is a subtlety in defining the inner product as in (2.21). This is a consequence of the fact that the partition function of Liouville CFT on the torus is strictly infinite due to the continuum of primary states. Indeed, as discussed around equations (2.28) and (2.32) from the definition of the inner product as an integral over Teichmüller space, the inner product between the torus characters is strictly speaking infinite. Nonetheless, there is a well-defined density of states in Liouville CFT obtained by dividing the partition function by an infinite overall prefactor. This density of states is uniform in the Liouville momentum

$$\rho_{\text{Liouville}}(P) = 1 \, , \tag{3.18}$$

and indeed in (2.32) we found that upon removing the infinite prefactors, the inner product between Virasoro characters is given by

$$\left\langle \mathcal{F}_{1,0}(P_1) \middle| \mathcal{F}_{1,0}(P_2) \right\rangle = \delta(P_1 - P_2) \, . \tag{3.19}$$

To summarize, in the case of a boundary torus without operator insertions, the Hilbert space is well-defined and although the inner product is formally infinite, it can be rendered sensible by removal of an overall infinite prefactor.

A technique for the computation of three-manifold invariants from TQFT state sums was pioneered by Turaev and Viro [93]. Here we briefly sketch the Turaev-Viro construction — closely following a nice explanation of the TQFT reasoning behind the prescription [94] — to illustrate why it cannot straightforwardly be applied in Virasoro TQFT. The construction works for any TQFT with gapped (i.e. topological) boundary conditions. In particular two copies of Virasoro TQFT have such a gapped boundary condition. Consider a three-manifold $M$ (say for simplicity without boundaries or Wilson lines), and equip it with a triangulation into tetrahedra.[26] The Turaev-Viro construction proceeds by carving out a three-ball in the vicinity of each vertex of the triangulation, which introduces $S^2$ boundaries at each vertex on which we impose the gapped boundary conditions. In ordinary compact TQFT, the partition function on the three-ball prepares a state in the one-dimensional Hilbert space associated with the boundary two-sphere. The TQFT path integral on this manifold is then equal to that

---

[26]It is a deep theorem that any three-manifold (possibly with boundary) can be triangulated [95, 96].

on $M$ up to a constant of proportionality $d$ that relates the state associated with each two-sphere boundary to that prepared by the three-ball for each vertex. $d^2$ is seen to be given by the partition function of the theory on the three-sphere, which follows since we can glue two three-balls together to form a three-sphere.

One can then continue and drill cylinders out between the three-balls, which prepare a state in the annulus Hilbert space. Continuing this drilling process leads to the following formula first conjectured by Turaev and Viro:

$$Z_{\text{TV}}(M) = \sum_{\text{decorations } \varphi} d^{-2\nu} \prod_{i=1}^{e} d_{\varphi(i)}^2 \prod_{a=1}^{t} |Z_{\varphi,a}|, \tag{3.20}$$

where $\nu$ is the number of vertices, $e$ is the number of edges, and $t$ is the number of three-simplices (tetrahedra) in the triangulation of $M$. The sum is over the possible ways of decorating each edge of the triangulation with Wilson lines in the spectrum of the TQFT. Here $d$ is the factor associated with the TQFT path integral with $S^2$ boundary for each vertex, $d_i$ is associated with the TQFT path integral on the solid cylinder with bulk Wilson line $L_i$, and $Z_{\varphi,a}$ is the TQFT path integral on the individual three-simplex $a$ with Wilson line edges specified by the decoration $\varphi$.[27] Turaev and Viro showed that if $d_i$ is given by the quantum dimension associated with the Wilson line $L_i$

$$d_i = \frac{\mathbb{S}_{\mathbb{1}i}[\mathbb{1}]}{\mathbb{S}_{\mathbb{1}\mathbb{1}}[\mathbb{1}]}. \tag{3.21}$$

$d$ is given by the "total quantum dimension"

$$d = \sqrt{\sum_i d_i^2}, \tag{3.22}$$

and $Z_{\varphi,a}$ is given by an appropriate $6j$ symbol, then $Z_{\text{TV}}(M)$ is invariant under the choice of triangulation.

At this point it should be clear that partition functions in $(\text{Virasoro TQFT})^2$ cannot straightforwardly be computed using the Turaev-Viro procedure. It fails at the very first step after triangulation of $M$, as the Hilbert space associated with a boundary two-sphere is not well-defined in Virasoro TQFT. Similarly, the quantum dimensions associated to Wilson lines in Virasoro TQFT are ill-defined precisely because the trivial line does not appear in the spectrum, so $\mathbb{S}_{\mathbb{1}\mathbb{1}}[\mathbb{1}]$ is not defined. One may try to consider a generalized notion of quantum dimension associated with the identity element of the Virasoro modular S-matrix

$$d_i \overset{?}{\propto} \mathbb{S}_{\mathbb{1}i}[\mathbb{1}] = \rho_0(P_i) = 4\sqrt{2}\sinh(2\pi b P_i)\sinh(2\pi b^{-1} P_i), \tag{3.23}$$

but even with this definition the analog of the total quantum dimension is clearly divergent

$$d^2 \overset{?}{\propto} \int_{\mathbb{R}} dP \rho_0(P)^2 = \infty. \tag{3.24}$$

The upshot of this discussion is that the most naive surgery techniques cannot directly be applied in the computation of Virasoro TQFT partition functions and so we will need another strategy. In the following subsections we describe techniques that facilitate the computation of Virasoro TQFT partition functions that only involve cutting three-manifolds along surfaces for which the TQFT Hilbert space is well-defined. We will provide significant evidence that at least in situations where the three-manifold is hyperbolic, these techniques yield finite results [52].

---

[27]Here boundary vertices and boundary edges should be counted with weight $\frac{1}{2}$ rather than 1.

## 3.4 Surface bundles over a circle

A simple class of computable examples is furnished by three-manifolds that are bundles of a surface $\Sigma_{g,n}$ over a circle. We can think of such surface bundles as a Euclidean wormhole $\Sigma_{g,n} \times [0,1]$ quotiented by the action of an element $\gamma$ of the boundary mapping class group $\mathrm{Map}(\Sigma_{g,n})$ that identifies the two boundaries of the wormhole

$$M_{g,n;\gamma} = \left(\Sigma_{g,n} \times [0,1]\right) / \{(x,0) \sim (\gamma(x),1)\}. \tag{3.25}$$

In many situations the resulting three-manifold $M_{g,n;\gamma}$ (which is sometimes referred to as a "mapping torus") is hyperbolic. Indeed, there is a theorem due to Thurston [97] that shows that $M_{g,n;\gamma}$ is hyperbolic if and only if $\gamma$ is what is known as a "pseudo-Anosov" homeomorphism of $\Sigma_{g,n}$.

We can then use the fact that the Virasoro TQFT is equipped with a (projective) representation of the boundary mapping class group $\mathrm{Map}(\Sigma_{g,n})$ on the Hilbert space $\mathcal{H}_{\Sigma_{g,n}}$ to straightforwardly compute the partition function on such surface bundles. Indeed, the TQFT partition function on the surface bundle is simply given by the trace of the operator $U(\gamma)$ in the boundary Hilbert space $\mathcal{H}_{\Sigma_{g,n}}$:

$$Z_{\mathrm{Vir}}(M_{g,n;\gamma}) = \mathrm{tr}_{\mathcal{H}_{\Sigma_{g,n}}} \left( U(\gamma) \right) \tag{3.26}$$

$$= \int \mathrm{d}^{3g-3+n} \vec{P} \, \rho^{\mathcal{C}}_{g,n}(\vec{P}) \left\langle \mathcal{F}^{\mathcal{C}}_{g,n}(\vec{P}) \middle| U(\gamma) \middle| \mathcal{F}^{\mathcal{C}}_{g,n}(\vec{P}) \right\rangle, \tag{3.27}$$

where $U(\gamma)$ is the representation of the modular transformation $\gamma$ on the Hilbert space $\mathcal{H}_{\Sigma_{g,n}}$. A priori, it is not clear when this trace is finite. It is obviously badly divergent when $\gamma$ corresponds to the identity since the Hilbert space is infinite-dimensional. Given that hyperbolic manifolds are the classical solutions of three-dimensional gravity, we expect that the trace (3.26) is finite in the case that the surface bundle $M_{g,n;\gamma}$ is hyperbolic. In other words, if $M_{g,n;\gamma}$ is hyperbolic, then $U(\gamma)$ should be a trace class operator on the Hilbert space $\mathcal{H}_{\Sigma_{g,n}}$.

As an example, consider the once-punctured torus fibered over the circle, pictured in figure 3. The mapping class group in this case is just $\mathrm{PSL}(2,\mathbb{Z})$, so the mapping tori are labeled by elements $\gamma \in \mathrm{PSL}(2,\mathbb{Z})$ (along with the deficit angle at the puncture, corresponding to the external operator dimension in the CFT). The hyperbolic mapping tori are hence specified by $\mathrm{PSL}(2,\mathbb{Z})$ elements $\gamma$ with two distinct real eigenvalues [98].

## 3.5 Heegaard splitting without boundaries

In order to compute partition functions of Virasoro TQFT on hyperbolic three-manifolds, we seek a cutting procedure that only introduces boundaries on which the Hilbert space of the TQFT is well-defined. A standard technique in three-dimensional topology is known as "Heegaard splitting" [99–101]. Consider a closed orientable three-manifold $M$ without any Wilson line insertions. $M$ is said to admit a "genus-$g$ Heegaard splitting" if it can be decomposed into two handlebodies identified along a common genus-$g$ boundary:

$$M = \mathrm{S}\Sigma_g^{(1)} \cup_\gamma \mathrm{S}\Sigma_g^{(2)}. \tag{3.28}$$

Here, $\mathrm{S}\Sigma_g$ are handlebodies, and $\gamma$ is an element of the mapping class group $\mathrm{Map}(\Sigma_g)$ of the boundary. The common boundary of the handlebodies

$$\partial(\mathrm{S}\Sigma_g^{(1)}) = \partial(\mathrm{S}\Sigma_g^{(2)}) = \Sigma_g, \tag{3.29}$$

is a Riemann surface known as the "splitting surface," and the boundaries are identified after the action of the modular transformation $\gamma$. An example is pictured in figure 4.

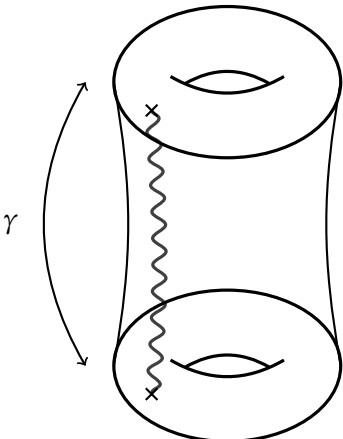

Figure 3: The once-punctured torus fibered over the circle. The mapping torus is specified by the strength of the puncture and the mapping class group element $\gamma \in \mathrm{PSL}(2, \mathbb{Z})$.

Every closed orientable three-manifold admits a Heegaard splitting, which follows easily from the triangulability of three-manifolds [95]. To see this, consider a triangulation of the three-manifold. A slight thickening of the one-skeleton of the triangulation defines a handlebody. The closure of the complement of this handlebody in fact defines another handlebody, associated with the thickening of the one-skeleton of the dual triangulation. This description is precisely the Heegaard splitting of the three-manifold as the union of handlebodies along their common boundary as in (3.28).

Given a Heegaard presentation of a three-manifold $M$ as in (3.28) it is straightforward to compute the Virasoro TQFT path integral on $M$. The TQFT path integral on the handlebody prepares a state in the Hilbert space associated with the splitting surface $\mathcal{H}_{\Sigma_g}$. In the absence of boundary Wilson line insertions, the boundary Hilbert space is well-defined in Virasoro TQFT provided the splitting surface has genus $g \geq 2$. In the absence of bulk Wilson lines, the state prepared by the TQFT path integral on the handlebody is simply given by the Virasoro identity block on $\Sigma_g$ in the channel $\mathcal{C}$ prescribed by the handlebody as in (3.10). The TQFT path integral on $M$ is then given by the matrix element of the representation $U(\gamma)$ of the mapping class group element $\gamma$ on the boundary Hilbert space $\mathcal{H}_{\Sigma_g}$ in the states prepared by the path integral on the handlebodies (3.10):

$$Z_{\mathrm{Vir}}(M) = \left\langle \mathcal{F}_g^{\mathcal{C}}(\vec{\mathbb{1}}) \,\middle|\, U(\gamma) \,\middle|\, \mathcal{F}_g^{\mathcal{C}}(\vec{\mathbb{1}}) \right\rangle . \tag{3.30}$$

As described in section 3.3, the identity block corresponds to a non-normalizable state in the boundary Hilbert space $\mathcal{H}_{\Sigma_g}$. So the matrix element (3.30) is not guaranteed to be finite; indeed for $\gamma$ equal to the identity it clearly diverges.

It is nontrivial to characterize the mapping class group elements $\gamma$ for which the matrix element (3.30) is finite; heuristically, it is finite for "sufficiently complex" $\gamma$. We conjecture that the TQFT partition function on $M$ is finite whenever $M$ is a hyperbolic three-manifold, since it

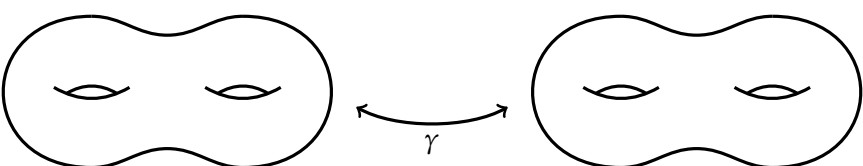

Figure 4: A genus-2 Heegaard splitting.

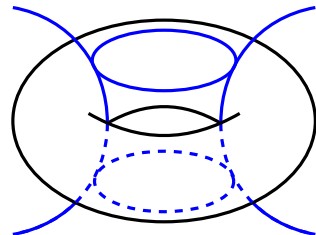

Figure 5: The Heegaard splitting of $S^3$ into two solid tori. We view $S^3$ as the one-point compactification of $\mathbb{R}^3$. One solid torus is drawn in black and the other complementary torus in blue, which in particular includes the compactification point.

is in those cases that AdS$_3$ gravity admits a classical solution and indeed a sufficiently complex $\gamma$ leads almost surely to a hyperbolic 3-manifold [79]. This expectation will be validated in the explicit examples we study in detail in a forthcoming companion paper [52]. However, there is no clear expectation on the converse statement – the inner product could be finite even though the manifold does not admit a hyperbolic metric.

As a standard example of the Heegaard splitting procedure consider the three-sphere $S^3$. Cutting the three-sphere along a torus, we obtain a Heegaard splitting of $S^3$ involving two solid tori — one with the $a$-cycle of the boundary torus filled in, the other with the $b$-cycle filled in — glued along their common boundary torus as pictured in figure 5. This is strictly speaking not well-defined in Virasoro TQFT because, as discussed in section 3.3, the Hilbert space on the torus without any boundary insertions is ill-defined due to the formally infinite inner product (2.32). Accepting that the path integral on the solid torus gives the Virasoro identity character, then from (3.30) we hence obtain the identity-identity component of the modular S-matrix

$$Z_{\text{Vir}}(S^3) \stackrel{?}{=} \mathbb{S}_{\mathbb{1}\mathbb{1}}[\mathbb{1}]. \tag{3.31}$$

Of course, as described in section 3.3, this is not well-defined in Virasoro TQFT since the spectrum does not contain the trivial line.

## 3.6 Heegaard splitting with boundaries

In the previous subsection we have shown how Heegaard splitting allows us to cut closed three-manifolds $M$ into handlebodies and how that decomposition allows a straightforward computation of partition functions of Virasoro TQFT on $M$ (given in (3.30)) by application of the inner product on the Hilbert space of the splitting surface. In order to compute TQFT partition functions on three-manifolds with boundaries, we require a cutting procedure that involves elements that themselves have boundaries beyond the splitting surface associated with the cutting. The required new concept is that of a "compression body," and the Heegaard splitting involving compression bodies rather than handlebodies is known in the literature as "generalized Heegaard splitting."

To develop the notion of a compression body, start by considering the handlebody $\mathsf{S}\Sigma_g$ with boundary $\Sigma_g$. We then imagine drilling out another handlebody in the interior of $\mathsf{S}\Sigma_g$; the resulting manifold (let's call it $W_g$) now has two $\Sigma_g$ boundaries, and is topologically equivalent to the Euclidean wormhole

$$W_g \simeq \Sigma_g \times [0,1]. \tag{3.32}$$

We then proceed to degenerate some cycles in the interior; we do this by attaching 2-handles along disjoint loops in the interior boundary, and fill in any resulting $S^2$ boundaries with three-balls.[28] The resulting three-manifold is known as a compression body $C_g(g_1, \ldots, g_m)$. See

---

[28]In the most general setting one need not fill in all the $S^2$ boundaries with three-balls. But for the purposes of

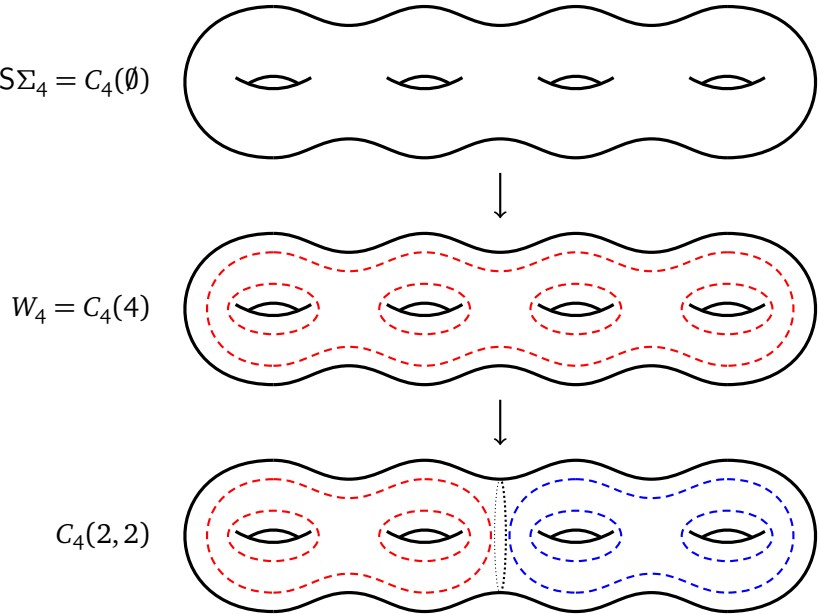

Figure 6: The construction of the $C_4(2,2)$ compression body. We start with the genus-4 handlebody $S\Sigma_4$, then drill out another genus-4 handlebody in the interior to form the genus-4 Euclidean wormhole $W_4 = C_4(4)$. We then degenerate a cycle in the interior boundary to form the compression body $C_4(2,2)$ with two genus-2 boundaries in the interior.

figure 6 for an example of a compression body with a genus-4 outer boundary and two genus-2 inner boundary components. We refer to the outer boundary $\Sigma_g \times \{0\}$ as $\partial_+C$ and the remaining (now possibly disconnected) boundary components $\partial C \backslash \partial_+ C$ as $\partial_- C$.

We have already discussed some trivial special cases of compression bodies. For example, if there are zero interior boundary components $\partial_- C = \emptyset$ ($m = 0$) then $C_g$ is simply a genus-$g$ handlebody. Similarly, if $\partial_- C$ has a single genus-$g$ component ($m = 1$, $g_1 = g$) then $C_g(g)$ is the Euclidean wormhole.

The triangulation theorem [95] for three-manifolds with boundaries shows that for any partition of the boundary components of a three manifold $M$, $\partial M = \partial M_1 \cup \partial M_2$, $M$ admits a Heegaard splitting involving two compression bodies $C^{(1)}$ and $C^{(2)}$ glued along their outer boundary, the splitting surface $\partial_+ C^{(1)} = \partial_+ C^{(2)} = \Sigma_g$

$$M = C_g^{(1)}(g_1, \ldots, g_{m_1}) \cup_\gamma C_g^{(2)}(h_1, \ldots, h_{m_2}), \tag{3.33}$$

with $\partial_- C^{(1)} = \partial M_1$, $\partial_- C^{(2)} = \partial M_2$ and $\gamma$ an element of $\text{Map}(\Sigma_g)$. Hence the Heegaard splitting may be regarded as a cobordism between the boundary components $\partial M_1$ and $\partial M_2$ [101].

To compute TQFT partition functions on three-manifolds $M$ with boundaries, we find a generalized Heegaard splitting of $M$ (3.33) and evaluate the matrix element of the representation of the mapping class group element $\gamma$ on the Hilbert space of the splitting surface $\mathcal{H}_{\Sigma_g}$ between the states prepared by the TQFT path integral on the compression bodies:.

$$Z_{\text{Vir}}(M) = \langle Z_{\text{Vir}}(C_g^{(1)}(g_1, \ldots, g_{m_1})) | U(\gamma) | Z_{\text{Vir}}(C_g^{(2)}(h_1, \ldots, h_{m_2})) \rangle. \tag{3.34}$$

We hence need to compute the partition function of Virasoro TQFT on compression bodies $C_g(g_1, \ldots, g_m)$. To do this, we insert a complete set of states (i.e. conformal blocks

---

computation of partition functions of Virasoro TQFT, they must be filled in since the Hilbert space on the sphere without sufficiently many Wilson line insertions is not well-defined.

$|\mathcal{F}_{g_i}^{\mathcal{C}_i}\rangle \in \mathcal{H}_{\Sigma_{g_i}})$ on the Hilbert spaces associated with each of the $m$ inner boundary components that make up $\partial_- C$. This prepares the following state

$$Z_{\text{Vir}}(C_g(g_1,\ldots,g_m)) = \int \prod_{i=1}^{m} \left(\mathrm{d}^{3g_i-3}\vec{P}_i \, \rho_{g_i}^{\mathcal{C}_i}(\vec{P}_i)\right)$$
$$\times \left|\mathcal{F}_{g_1}^{\mathcal{C}_1}(\vec{P}_1)\right\rangle \otimes \cdots \otimes \left|\mathcal{F}_{g_m}^{\mathcal{C}_m}(\vec{P}_m)\right\rangle \otimes \left|\Psi_g^{\mathcal{C}}(\vec{P}_1,\ldots,\vec{P}_m)\right\rangle. \quad (3.35)$$

For each complete set of states in the inner boundary Hilbert spaces there is the freedom of a choice of basis, in particular to specify a channel $\mathcal{C}_i$ in the conformal block decomposition. The state $\left|\Psi_g^{\mathcal{C}}(\vec{P}_1,\ldots\vec{P}_m)\right\rangle \in \mathcal{H}_{\Sigma_g}$ is a particular genus-$g$ conformal block associated with the outer boundary $\partial_+ C = \Sigma_g$. It is obtained by gluing the handlebodies with Wilson lines corresponding to the conformal blocks appearing in the complete set of states in the inner boundary Hilbert spaces into the compression body. The resulting $\Sigma_g$ conformal block is hence made up of sub-blocks that are identical to those of the inner boundaries, with paired conformal weights propagating in the same $\mathcal{C}_i$ sub-channels, and only the identity operator propagating through disks that have been degenerated in the inner boundary. See figure 7 for an example.

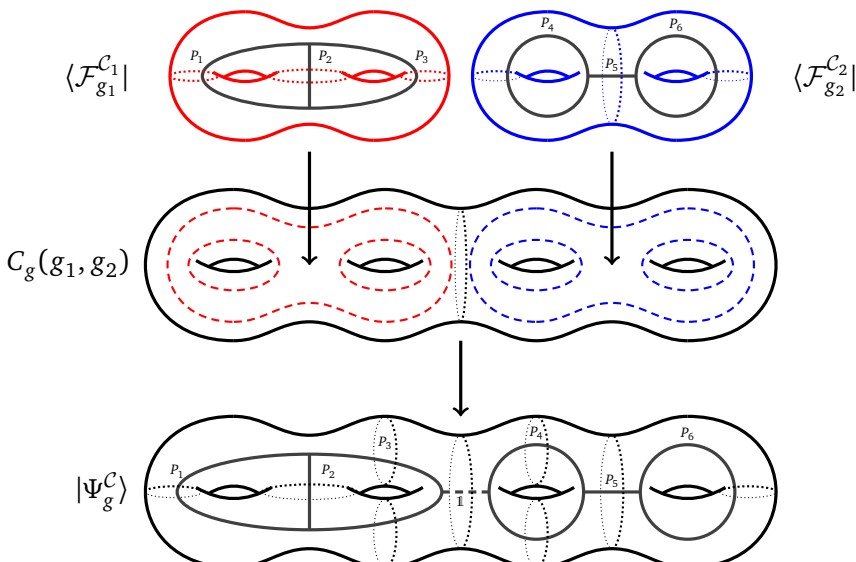

Figure 7: When inserting a complete set of states in the Hilbert space of the inner boundaries, we glue handlebodies with Wilson lines corresponding to conformal blocks of the inner boundaries into the compression body. This prepares a particular conformal block in the Hilbert space of the outer boundary. Only the identity can propagate through any remaining empty disks.

The compression body partition function is then a sort of generalization of the Liouville partition function, in that it involves integrated conformal blocks associated with the outer and inner boundaries with paired internal conformal weights (but now the conformal blocks are associated with different surfaces). These generalized Liouville partition functions are invariant with respect to the bulk mapping class group of the compression body. This invariance generalizes the fact that the wormhole partition function (3.4) is invariant under the diagonal mapping class group that acts simultaneously on the two boundaries.

For concreteness, consider for example the compression body $C_4(2,2)$ depicted in figure 6. Inserting a complete set of states in the two genus-two Hilbert spaces of the inner boundaries,

we have

$$Z_{\text{Vir}}(C_4(2,2)) = \int \prod_{i=1}^{6} \left( \mathrm{d}P_i \, \rho_0(P_i) \right) C_0(P_1, P_2, P_3)^2 C_0(P_4, P_4, P_5) C_0(P_5, P_6, P_6)$$

$$\times \Big| \quad \quad \quad \quad \quad \quad \quad \quad \quad \quad \quad \quad \quad \quad \quad \quad \Big\rangle.$$

(3.36)

### 3.7 Heegaard splitting with Wilson lines and/or boundaries

In order to compute partition functions of Virasoro TQFT on three-manifolds with boundary Wilson line insertions, we need to further generalize the Heegaard splitting procedure. Although conceptually the strategy is exactly the same as without Wilson lines — in particular, to cut the manifold into generalizations of compression bodies, and then to compute the path integral on these building blocks by resolving the identity on the Hilbert spaces associated with the inner boundaries — we will see that the introduction of Wilson lines leads to some complications that need to be treated carefully.

We now consider cutting the three-manifold along a splitting surface that is a punctured Riemann surface $\Sigma_{g,n}$. This decomposes the three-manifold into a union of generalized compression bodies with Wilson lines

$$M = C_{g,n}^{(1)}(g_1, n_1; \ldots; g_{m_1}, n_{m_1}) \cup_\gamma C_{g,n}^{(2)}(h_1, p_1; \ldots; h_{m_2}, p_{m_2}). \tag{3.37}$$

The generalized compression bodies $C_{g,n}(g_1, n_1; \ldots; g_m, n_m)$ are compression bodies with Wilson lines connecting the inner and outer boundaries (and possibly connecting distinct inner boundaries), arranged so that

$$\partial_+ C = \Sigma_{g,n}, \quad \partial_- C = \bigsqcup_{i=1}^{m} \Sigma_{g_i, n_i}. \tag{3.38}$$

Our notation does not keep track of the precise shape of the Wilson lines. Note that unlike the case without Wilson lines, we can now consider the situation that $\partial_- C$ contains sphere components — provided that they are supported by at least three Wilson line insertions — in Virasoro TQFT, since the associated Hilbert spaces are well-defined. See figure 8 for an example.

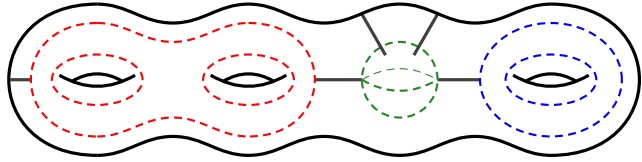

Figure 8: The pictured compression body $C_{3,3}(2, 2; 0, 4; 1, 1)$ involves Wilson lines connecting the outer boundary with the inner boundaries and the inner boundaries to each other.

The computation of the TQFT partition function on the generalized compression bodies proceeds in exactly the same way as on the compression bodies without Wilson line insertions. We insert a complete set of states in the Hilbert spaces $\mathcal{H}_{\Sigma_{g_i, n_i}}$ associated with the inner boundaries, which expresses the partition function as a linear combination of products of conformal blocks of each of the boundaries. As without Wilson lines the conformal block associated with

the outer boundary has the identity operator propagating in all channels corresponding to cycles that have been degenerated in forming the inner boundaries. This prepares a state very similar to (3.35) except that the conformal blocks that appear in the complete sets of states may have external punctures

$$Z_{\text{Vir}}(C_{g,n}(g_1, n_1; \ldots; g_m, n_m)) = \int \prod_{i=1}^{m} \left( \mathrm{d}^{3g_i - 3 + n_i} \vec{P}_i \, \rho_{g_i, n_i}^{C_i}(\vec{P}_1) \right)$$
$$\times \left| \mathcal{F}_{g_1, n_1}^{C_1}(\vec{P}_1) \right\rangle \otimes \cdots \otimes \left| \mathcal{F}_{g_m, n_m}^{C_m}(\vec{P}_m) \right\rangle \otimes \left| \Psi_{g,n}^{C}(\vec{P}_1, \ldots, \vec{P}_m) \right\rangle.$$
$$(3.39)$$

The partition function on the Euclidean wormhole (3.1) is a trivial example of a compression body partition function, with $m = 1$ and $(g_1, n_1) = (g, n)$.

In the presence of Wilson lines, the state $\left| \Psi_{g,n}^{C}(\vec{P}_1, \ldots, \vec{P}_m) \right\rangle \in \mathcal{H}_{\Sigma_{g,n}}$ prepared by this procedure — namely, by gluing into the compression body the punctured handlebodies with bulk Wilson lines corresponding to the conformal blocks appearing in the complete set of states — may be somewhat complicated. For example, it may be the case that Wilson lines connecting boundaries of the generalized compression body are braided or knotted in the bulk. It may be possible to straighten the Wilson lines by braiding the operator insertions on one of the boundaries (i.e. by sequential application of (2.36) to external legs of one of the boundary conformal blocks), but this is not always possible. For example, it may be that the Wilson lines trace out a "non-rational tangle" as in figure 9 in the bulk (see [102, 103] for related discussions in the context of 3d gravity). In this case, we can perform a further splitting by cutting through the non-rational tangle in the bulk along a surface with more Wilson line insertions, and then performing crossing transformations on this surface to straighten the Wilson lines. We will discuss the case of non-rational tangles in more explicit detail in [52].

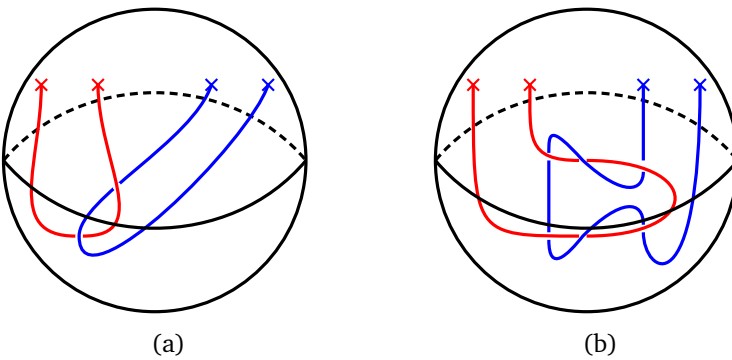

(a)                    (b)

Figure 9: An example of (a) rational and (b) non-rational tangles of Wilson lines that might appear as states in the Hilbert space of the sphere with four punctures.

Another potential subtlety is that as a result of projecting the compression body onto the inner boundary conformal blocks, the resulting configuration may involve loops of Wilson lines in the bulk, for which the interpretation in terms of conformal blocks in the Hilbert space of the splitting surface is unclear. To illustrate the idea, consider the following generalized

compression body:

$$M = \qquad\qquad\qquad\qquad\qquad\qquad \text{(3.40)}$$

This is a Euclidean wormhole with four asymptotic boundaries, each of which is a sphere with three operator insertions. This four-boundary wormhole controls the leading non-Gaussianities in the large-$c$ ensemble of CFT data dual to semiclassical 3d gravity [15]. The partition function on the four-boundary wormhole can be straightforwardly computed by Heegaard splitting as described in this section as[29]

$$Z_{\mathrm{Vir}}(M) = \rho_0(P_t)^{-1} C_0(P_1, P_2, P_s) C_0(P_3, P_4, P_s) \mathbb{F}_{P_s P_t} \begin{bmatrix} P_1 & P_4 \\ P_2 & P_3 \end{bmatrix}$$

$$\times \left| \begin{array}{cccc} {}^1 \!\!\bigotimes\!\! {}^2 & {}^1 \!\!\bigotimes\!\! {}^t & {}^t \!\!\bigotimes\!\! {}^2 & {}^4 \!\!\bigotimes\!\! {}^3 \\ {}_s & {}_4 & {}_3 & {}_s \end{array} \right\rangle, \qquad\qquad \text{(3.41a)}$$

$$= \rho_0(P_s)^{-1} C_0(P_1, P_4, P_t) C_0(P_2, P_3, P_t) \mathbb{F}_{P_t P_s} \begin{bmatrix} P_1 & P_2 \\ P_4 & P_3 \end{bmatrix}$$

$$\times \left| \begin{array}{cccc} {}^1 \!\!\bigotimes\!\! {}^2 & {}^1 \!\!\bigotimes\!\! {}^t & {}^t \!\!\bigotimes\!\! {}^2 & {}^4 \!\!\bigotimes\!\! {}^3 \\ {}_s & {}_4 & {}_3 & {}_s \end{array} \right\rangle. \qquad\qquad \text{(3.41b)}$$

We have included the kets on the right-hand side to remind the reader that this is a state in the tensor product of the Hilbert spaces associated with each asymptotic boundary $Z_{\mathrm{Vir}}(M) \in \mathcal{H}_{\Sigma_{0,3}}^{\otimes 4}$, even though the Hilbert spaces in this case are one-dimensional. As discussed in section 3.2, we are adopting conventions such that the trivalent junction of Wilson lines is unit-normalized. It is a nontrivial fact that the four-boundary wormhole partition function (3.41) enjoys a tetrahedral symmetry permuting the Wilson lines from the bulk topology, though it is not at all evident from this presentation. Indeed, this combination is proportional to the $6j$ symbol of the modular double $U_q(\mathfrak{sl}(2, \mathbb{R}))$ [49, 50]. As already remarked in section 2.5 around eq. (2.52), that this combination has tetrahedral symmetry is a consequence of the pentagon identity (A.4) satisfied by the fusion kernel, but one may also derive it by computing the partition function of the TQFT on the four-boundary wormhole through different Heegaard splittings. More examples of miraculous-looking identities for the crossing kernels and conformal blocks that can be derived through consistency of the TQFT will be discussed in [52].

We could also compute the partition function of this compression body by trivially resolving the identity on the Hilbert spaces of the three inner boundaries. This prepares a state with a

---

[29]In [52] we provide more details of the computation of this partition function and discuss in more detail the statistical interpretation of this four-boundary wormhole in the ensemble dual of semiclassical 3d gravity.

loop of Wilson lines in the Hilbert space of the outer boundary:

$$Z_{\text{Vir}}(M) = C_0(P_1, P_4, P_t)C_0(P_2, P_3, P_t)C_0(P_3, P_4, P_s) \left| \begin{array}{c} \end{array} \right\rangle$$

$$\times \left| \begin{array}{ccc} \end{array} \right\rangle . \tag{3.42}$$

By comparing with the computation of the partition function of the four-boundary wormhole via Heegaard splitting (3.41), we can determine the normalization of the state on the outer boundary involving the Wilon line loop as:

$$\left| \begin{array}{c} \end{array} \right\rangle = \frac{C_0(P_1, P_2, P_s)\mathbb{F}_{P_s P_t}\begin{bmatrix} P_1 & P_4 \\ P_2 & P_3 \end{bmatrix}}{\rho_0(P_t)C_0(P_1, P_4, P_t)C_0(P_2, P_3, P_t)} \left| \begin{array}{c} \end{array} \right\rangle \tag{3.43}$$

$$= \frac{C_0(P_1, P_2, P_s)\mathbb{F}_{P_t P_s}\begin{bmatrix} P_1 & P_2 \\ P_4 & P_3 \end{bmatrix}}{\rho_0(P_s)C_0(P_1, P_2, P_s)C_0(P_3, P_4, P_s)} \left| \begin{array}{c} \end{array} \right\rangle . \tag{3.44}$$

More complicated loop configurations of Wilson lines (for example involving more Wilson lines attached to the loop, or more internal loops) can always be reduced to simple loops of this kind by applying crossing transformations, so it suffices to work out this identity.

To make contact with a familiar identity from more conventional TQFTs, it is instructive to consider the limit in which the $s$ Wilson line becomes the trivial line (i.e. $P_s \to iQ/2$). Comparing the corresponding limits of (3.41) and (3.42), we conclude that a Wilson line bubble can be reduced to a single Wilson line as follows

$$1 \!\!-\!\!\bigcirc\!\!-\!\! 2 \;=\; \frac{\delta(P_1 - P_2)}{\rho_0(P_1)C_0(P_1, P_3, P_4)} \;\; 1 \tag{3.45}$$

Here we've replaced $P_t \to P_4$ for better readability.

In any case, we can always reduce the three-manifold to a union of generalized compression bodies involving only unknotted Wilson lines connecting the various boundaries. For these configurations, the evaluation of the compression body partition function as in (3.39) is straightforward by insertion of complete sets of states in the Hilbert spaces of the inner boundaries.

# 4 Discussion

In this paper, we developed a systematic algorithm to compute partition functions of 3d gravity on arbitrary hyperbolic manifolds. We established the general theory in this paper and will demonstrate its usefulness by working out a number of explicit examples in a follow-up paper [52]. We end with a discussion of possible issues with our proposal, its relation to other literature and comment on future applications.

**Finiteness of the partition function.** Since the Hilbert space of Virasoro TQFT is infinite-dimensional, not all partition functions are finite. One expects on physical grounds that $Z_{\mathrm{Vir}}(M)$ is at least well-defined for all hyperbolic manifolds. While we observe this to be the case in all examples that we considered, we do not have a general argument to prove this statement.

It is conceivable that $Z_{\mathrm{Vir}}(M)$ is still well-defined for some non-hyperbolic manifold, although we are not aware of an example.[30] Even if $Z_{\mathrm{Vir}}(M)$ is divergent, one might hope that the gravity partition function is rendered finite by dividing by the bulk mapping class group $\mathrm{Map}_0(M, \partial M)$, which does not have to be finite for non-hyperbolic manifolds. This certainly does not work in general ($\Sigma \times S^1$ or trivially $S^3$ are counterexamples). However, it was argued in [11] that Seifert manifolds (which are in general non-hyperbolic) can give finite contributions to the gravitational path integral. They admit a simple Heegaard splitting [104] and thus provide an interesting arena for further study of off-shell contributions to the gravitational path integral.

**Volume conjecture and refined volume conjecture.** For hyperbolic manifolds, we can compare our computation to the one-loop evaluation of the gravitational path integral. Using the results of [105] for the one-loop determinant, we hence suspect that

$$|Z_{\mathrm{Vir}}(M)|^2 = e^{-\frac{c}{6\pi}\mathrm{vol}(M)}\left[\prod_{\gamma \in \mathcal{P}}\prod_{m=2}^{\infty}\frac{1}{|1-q_\gamma^m|^2} + \mathcal{O}(c^{-1})\right]. \tag{4.1}$$

Here, $\mathcal{P}$ denotes the set of all primitive simple closed geodesics and $q_\gamma = e^{-\ell_\gamma}$, where $\ell_\gamma$ is the complex length of the geodesic. Equivalently, $\mathcal{P}$ is the set of all primitive conjugacy classes of the Kleinian group realizing the manifold $M$ and $q_\gamma^{1/2}$ is the smaller of the two eigenvalues of the corresponding $\mathrm{PSL}(2, \mathbb{C})$ matrix.[31] This formula is derived by using essentially the same logic as for the Selberg trace formula.

The leading behaviour in terms of the volume of the hyperbolic manifold is often discussed in the literature under the name of the volume conjecture, although it is often formulated for the analytically continued $\mathrm{SU}(2)_k$ Chern-Simons partition function [73–75, 106]. There is a refinement for $Z_{\mathrm{Vir}}(M)$ itself, where the imaginary part is given by the Chern-Simons invariant of the manifold [107]. Equivalently, the volume $\mathrm{vol}(M)$ is sometimes given as a complex number, e.g. by using the command `complex_volume()` in `SnapPy` [108].

Eq. (4.1) should also hold in the presence of boundaries, in which case $\mathrm{vol}(M)$ denotes the renormalized volume [82, 83, 109]. The renormalized volume has the same ambiguities in its definition as the Virasoro TQFT partition function which are controlled by the conformal anomaly.

To our knowledge, the "refined volume conjecture" given by eq. (4.1) and the case with boundaries has not been discussed in the literature before. It leads to very non-trivial predictions, e.g. about the semi-classical expansion of conformal blocks and it definitely seems worthwhile to explore this conjecture mathematically.

**Equivalence to Andersen and Kashaev.** A similar TQFT under the name of "Teichmüller TQFT" was originally proposed by Andersen and Kashaev [31, 32]. It is also based on the quantization of Teichmüller space. Their formalism is very different and relies on the quantization of Teichüller space in explicit coordinates (so-called Penner coordinates), where both

---

[30]There are some "almost" examples such as the unknot and the torus Euclidean wormhole that give well-defined answers up to ill-defined overall prefactors.

[31]We count the conjugacy class of $g$ and $g^{-1}$ only once since they lead to the same geodesic. This is the reason for an extra square compared to [105].

the symplectic form and the mapping class group transformations take a relatively simple form, which was developed in [30, 40, 41]. This quantization is not obviously related to the quantization in terms of conformal blocks, which is physically much more desirable. Thus, it is not obvious that the Virasoro TQFT discussed in this paper is equivalent to Teichmüller TQFT, although it is strongly suggested by various results in the literature [30, 110].

**State-sum models.** One initial motivation for the current study was to formulate 3d quantum gravity as a state-sum model in the same way as 2d gravity can be obtained from a scaling limit of random triangulations of surfaces [111–113]. Such attempts at state-sum models for 3d gravity were made in the past, the most well-known being the Ponzano-Regge model [114–121]. However, they are all somewhat ad-hoc constructions and essentially the Turaev-Viro construction of $\mathrm{SU}(2)_k \times \mathrm{SU}(2)_k$ Chern-Simons theory [93], which clearly does not adequately describe quantum gravity for the reasons discussed in this paper. As discussed in section 3.3, the Turaev-Viro construction does not carry over to $\mathrm{SL}(2, \mathbb{R})_k \times \mathrm{SL}(2, \mathbb{R})_k$ Chern-Simons theory because the total quantum dimension is ill-defined.

There is a restricted state-sum model of Teichmüller TQFT proposed by Andersen and Kashaev [31,32], where the authors only consider certain triangulations with additional structure (so-called shaped triangulations). In particular, the state-sum is only invariant under the so-called 2-3 Pachner move that keeps the number of vertices of the triangulation invariant, but not under the 1-4 Pachner move. Related constructions have appeared in the physics literature [122,123]. The relation of this restricted state-sum model to the construction discussed in this paper remains to be elucidated.

**Summing over topologies.** To define a full theory of quantum gravity, we should also sum the gravity partition function of a fixed topology eq. (1.1) over all topologies. Of course, there are a number of problems when performing this sum. While a sum over all three-manifolds might sound daunting, we shall now argue that the associated difficulties can optimistically be overcome.

An obvious problem is the existence of topologies with ill-defined partition function, such as the $S^3$ partition function. However, this should not be too worrying since a similar phenomenon also appears in two-dimensional gravity, where the sphere and the torus partition function are ambiguous or infinite [124–127]. This is mirrored by certain corresponding ambiguities in the dual matrix integral. We are thus motivated to discard such divergent contributions in three-dimensional gravity in an ad-hoc manner, since they are presumably mirrored by corresponding divergences in an ensemble boundary CFT interpretation. As already mentioned, there are several indications that in a certain sense almost all three-manifolds are hyperbolic and hence lead to finite contributions [76–79]. We can thus somewhat artificially restrict the sum to hyperbolic manifolds:

$$Z_{\mathrm{grav}}(\partial M) \overset{?}{=} \sum_{\text{hyperbolic manifolds } M} Z_{\mathrm{grav}}(M). \tag{4.2}$$

Here the notation indicates that the actual gravity partition function only depends on the boundary geometry. In the sum, we are summing over all hyperbolic manifolds compatible with the boundary conditions.

Let us discuss the case without boundaries, since the mathematics behind it are best understood. In this case, the size of the individual terms is to leading order in the semi-classical $\frac{1}{c}$ expansion given by $e^{-\frac{c}{6\pi}\mathrm{vol}(M)}$, see eq. (4.1). Thanks to the rigidity of hyperbolic 3-manifolds, closed hyperbolic 3-manifolds can be ordered according to their volume and $\mathrm{vol}(M)$ is a topological invariant playing a loose analogue of the genus in 2d gravity. For large central charges,

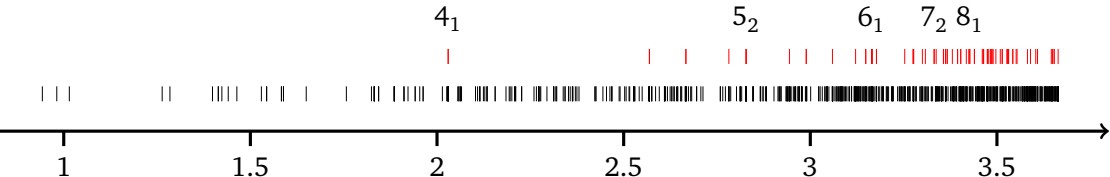

Figure 10: The volume spectrum of complete hyperbolic manifolds with finite volume below 3.6639. This list is taken from the Hodgson-Weeks census and might not be complete. We also plotted the spectrum of accumulation points above in red and identified some of the accumulation points as well-known knot-complements in $S^3$. The notation of the knots is standard and denotes the minimal number of crossings.

manifolds with large volumes are greatly suppressed. The volume spectrum of complete hyperbolic 3-manifolds of finite volume is a closed subset of $\mathbb{R}_{>0}$ of order type $\omega^\omega$ [76].[32] This means that there is a unique hyperbolic manifold of smallest volume (the so-called Weeks manifold of volume $\approx 0.9427$ [128]). However, the spectrum has accumulation points from below, the first accumulation point occurring at $\approx 2.0299$, corresponding to the complement of the figure 8 knot (a cusped hyperbolic 3-manifold) [129]. The first limit point of limit points occurs at volume $\approx 3.6639$ corresponding to the complement Whitehead link (a hyperbolic manifold with two cusps) [130]. Accumulation points occur because one can produce a family of hyperbolic 3-manifolds by performing a so-called Dehn filling of a cusp in a cusped hyperbolic manifold which we discuss further below. However, a lot of the structure of the volume spectrum is not well-understood. We plotted the volume spectrum of hyperbolic manifolds in the Hogdson-Weeks census of closed hyperbolic manifolds [131] in Figure 10 to give an idea of the structure.

There are hence two main issues: (1) the sum diverges because the density of hyperbolic manifolds presumably grows very fast and (2) the sum diverges because of the existence of accumulation points. To make (1) more precise, one should also study how the functional determinant behaves for different hyperbolic manifolds. On general grounds one expects that (4.2) is only an asymptotic sum. The same happens also already in two-dimensional models such as JT gravity [2, 132], where the asymptotic nature indicates the necessity to include non-perturbative (or "doubly-non-perturbative") objects corresponding to ZZ-instantons in the worldsheet description. Thus the fact that (4.2) is presumably an asymptotic sum indicates the presence of non-perturbative contributions to (4.2). It would be interesting to get a handle on the asymptotic nature of the sum (4.2) which would give non-trivial information about the structure of non-perturbative objects in the theory by resurgence analysis.[33]

To understand the issue (2) of accumulation points, we have to explain the procedure of Dehn filling in some more detail. Given a hyperbolic manifold $M$ with a cusp, we can excise a tubular neighborhood around the cusp and obtain a manifold with a boundary $\mathring{M}$. We can then glue back a solid torus via a mapping class group element $\gamma \in \mathrm{SL}(2, \mathbb{Z})/\mathbb{Z}$ to obtain a new manifold $M_\gamma$. Except for finitely many exceptions, $M_\gamma$ is hyperbolic and has lower volume than $M$ [76]. This is a splitting as described in the main text and hence the TQFT partition function can be computed as

$$\langle Z_{\mathrm{Vir}}(\mathring{M}) | U(\gamma) | \mathcal{F}_{1,0}(\mathbb{1}) \rangle, \tag{4.3}$$

where $\gamma \in \mathrm{PSL}(2, \mathbb{Z})/\mathbb{Z}$ and $U(\gamma)$ is the corresponding representation on the Hilbert space.

---

[32]In this set of manifolds, we do not require the manifolds to be compact and allow for the existence of cusps, but no conical defects.

[33]Such contributions would presumably correspond to doubly non-perturbative effects from the perspective of the $1/c$ expansion. That the $1/c$ expansion around a given bulk saddle is asymptotic may be viewed as indicating the need for the sum over geometries itself [133].

Thus the sum over topologies is locally labelled by an element in $\mathrm{PSL}(2, \mathbb{Z})/\mathbb{Z}$ near the a single accumulation point. To perform the sum over topologies obtained by a Dehn filling from a cusp, we hence merely need to compute

$$\sum_{\gamma \in \mathrm{PSL}(2,\mathbb{Z})/\mathbb{Z}} U(\gamma)|\mathcal{F}_{1,0}(\mathbb{1})\rangle \otimes \overline{U(\gamma)|\mathcal{F}_{1,0}(\mathbb{1})\rangle}. \tag{4.4}$$

This is however nothing other than the Maloney-Witten sum obtained for the thermal AdS$_3$ partition function [9]. Thus we conclude that the problem of accumulation points is the same as making sense of the Maloney-Witten sum.[34] It is known how to regularize the Maloney-Witten sum [9, 20], although problems with a small negativity of states have so far resisted a full explanation [11, 12, 22]. However, it should be clear from this discussion that it is not completely hopeless (although technically daunting) to make sense of the sum over topologies in 3d quantum gravity.

**Relation to CFT ensemble.** It has recently been proposed that semiclassical AdS$_3$ Einstein gravity coupled to point particles is dual to an ensemble of large-$c$ 2d CFTs with random structure constants [15]. This proposal was developed by formulating a Gaussian ansatz for the OPE coefficients, averaging products of CFT observables subject to this random ansatz together with an averaged spectrum given by the Cardy formula, and matching to on-shell actions (and one-loop determinants) of the corresponding classical solutions of Einstein's equations in the large-$c$ limit. While the averaged CFT computations are straightforward, the correspondence with semiclassical gravity was a nontrivial output of delicate gravity calculations. Our formulation of 3d gravity in terms of Virasoro TQFT trivializes or simplifies many computations that are burdensome or not tractable in the metric formalism, and in many cases extends the correspondence between averaged CFT observables and gravity path integrals on a given topology to finite central charge; this suggests that the match between gravity and averaged CFT observables is in an appropriate sense exact at the level of fixed topologies. In particular, the relation between the gravity path integral on the Euclidean wormhole $\Sigma \times [0, 1]$ and the product of Liouville CFT partition functions originally derived semiclassically in [15] follows straightforwardly from the resolution of the identity in the Hilbert space of the asymptotic boundaries as described in section 3.1.

While the large-$c$ ensemble of CFT data is crossing invariant on average, it is clear from a variety of points of view that solving the crossing equations requires non-Gaussian corrections to the statistics of the structure constants [14, 15, 135–138]. It is however unclear how to treat wormholes with more than two asymptotic boundaries that would encode such statistics in the metric formalism except in certain special situations. The Virasoro TQFT description of 3d gravity completely systematizes the computation of the gravity path integral on general multi-boundary wormholes with arbitrary configurations of Wilson lines, which in turn unambiguously determine higher moments of the CFT structure constants. The explicit example of the partition function of the four-boundary sphere three-point wormhole (3.42) that controls the non-Gaussian fourth moment of the structure constants was briefly discussed in section 3.7; this and more interesting examples will be discussed in more detail in [52].

It would be desirable to prove in full generality that the gravity path integral on a fixed topology agrees with the corresponding averaged CFT observable. While the Virasoro TQFT formulation facilitates the algorithmic computation of higher moments of OPE coefficients, we seek a unifying description of the boundary theory that captures all corrections to the Gaussian ensemble. Presumably, fully solving the boundary theory requires reckoning more

---

[34]There can be a finite number ($\leq 10$ [134]) of exceptional $\gamma$'s for which the resulting manifold is not hyperbolic. These have to be omitted in the sum if we only want to sum over hyperbolic manifolds.

seriously with the sum over topologies in the bulk, and perhaps with the path integral on off-shell toplogies. We leave a more comprehensive discussion of multi-boundary wormholes and the implications for the CFT ensemble to future work.

**Supergravity and higher spin gravity.** It is well-known that the Chern-Simons formalism can in principle be adapted to incorporate supersymmetry and/or higher spin fields by changing the gauge group to a (super)group G together with an embedding $\mathrm{SL}(2,\mathbb{R}) \longrightarrow$ G [139]. For example, for G = $\mathrm{OSp}(1|1,1)$, we get $\mathcal{N} = 1$ supergravity, for G = $\mathrm{OSp}(2|1,1)$ we get $\mathcal{N} = 2$ supergravity, while for G = $\mathrm{SL}(N,\mathbb{R})$ with the principal embedding, we get higher spin gravity with fields of spin $2, 3, \ldots, N$. In all these cases, there is a "Teichmüller component" of the phase space as studied in higher Teichmüller theory. The natural conjecture for the quantization of these spaces is in terms of $\mathcal{N} = 1$ Virasoro conformal blocks, $\mathcal{N} = 2$ Virasoro conformal blocks and $\mathcal{W}_N$ conformal blocks, respectively. There is a conceptual difficulty in the definition of $\mathcal{W}_N$ conformal blocks since not all descendant correlation functions can be reduced to primary correlation functions [140]. It stands to reason that at least in the case without higher spin fields one can similarly define a (spin) TQFT based on the crossing symmetry of super Virasoro blocks, see [141–143] for progress in this direction. Except for the $\mathcal{N} = 1$ Virasoro case, the existence of this TQFT has not been established. In particular the generalizations of the crossing kernels $\mathbb{F}$ and $\mathbb{S}$ are only known in the $\mathcal{N} = 1$ Virasoro case [144–147].

**Zero and positive cosmological constant.** There is an obvious question whether an analogous story exists for pure quantum gravity with $\Lambda = 0$ or $\Lambda > 0$, which is loosely related to Chern Simons theory with the Poincaré group or $\mathrm{SL}(2,\mathbb{C})$ as gauge group. The quantization of $\mathrm{SL}(2,\mathbb{C})$-Chern-Simons theory is actually better understood than for $\mathrm{SL}(2,\mathbb{R}) \times \mathrm{SL}(2,\mathbb{R})$ Chern-Simons theory [148], but one faces myriads of puzzles, see e.g. [36, 149–156].

**Engineering bulk duals to arbitrary CFTs.** It was recently suggested [157] that one can construct a toy model of holography by considering $\mathrm{SU}(2)_k \times \mathrm{SU}(2)_k$ Chern-Simons theory and gauging the non-invertible one-form symmetry generated by the diagonal lines. This leads to a theory that is trivial in the bulk (since it does not have any non-trivial gauge-invariant operators) and hence is tautologically dual to the $\mathrm{SU}(2)_k$ WZW model. In particular this TQFT is completely insensitive to the bulk topology and thus does not require a sum over different topologies. Modulo technical difficulties related to the continuum of lines, it should be possible to repeat this exercise for two copies of Virasoro TQFT. By gauging the set of all diagonal (i.e. non-spinning) lines, one can produce a bulk dual of Liouville theory.[35] In fact, one can in principle produce a tautological AdS/CFT pair for any CFT by gauging the set of lines corresponding to the operator spectrum of the CFT of interest. Consistency of the gauging should directly translate into the crossing equations of the boundary CFT. Gauging a set of lines requires one to insert a mesh of Wilson lines in the bulk and sum over all such insertions. This can be interpreted as the sum over topologies on the bulk gravity side. Inserting Wilson lines in Virasoro TQFT corresponds to putting conical defects or black hole horizons in the bulk. Thus such an artificially engineered bulk theory requires one to include such singular geometries and black hole geometries with precisely the right multiplicity in order to reproduce the boundary partition functions.

---

[35]The properties of such a theory were analyzed from the boundary side in [158].

## Acknowledgments

We thank Jan de Boer, Jeevan Chandra, Gabriele di Ubaldo, Henriette Elvang, Tom Hartman, Aidan Herderschee, Juan Maldacena, Alex Maloney, Jake McNamara, Baur Mukhametzhanov, Vladimir Narovlansky, Eric Perlmutter, Massimo Porrati, Guillaume Remy, Sylvain Ribault, Nati Seiberg, Sahand Seifnashri, Al Shapere, Douglas Stanford, Xin Sun, Ioannis Tsiares, Joaquin Turiaci, Erik Verlinde, Herman Verlinde, Yifan Wang, Gabriel Wong, Edward Witten and Sasha Zhiboedov for useful discussions. We especially thank Vladimir Narovlansky, Joaquin Turiaci and Herman Verlinde for initial collaboration.

**Funding information** S.C. is supported by the Sam B. Treiman fellowship at the Princeton Center for Theoretical Science. L.E. is supported by the grant DE-SC0009988 from the U.S. Department of Energy. S.C. and L.E. thank the organizers of the KITP workshop "Bootstrapping Quantum Gravity" for hopsitality during the course of this work. This research was supported in part by the National Science Foundation under Grant No. NSF PHY-1748958.

## A  Consistency conditions of the Moore-Seiberg construction

In this appendix, we list all the consistency conditions that the crossing kernels $\mathbb{F}$, $\mathbb{S}$ have to satisfy. They are defined by eqs. (2.42a) and (2.42b). $\mathbb{F}$ has the following symmetry properties:

$$\mathbb{F}_{P_{21},P_{32}}\begin{bmatrix} P_3 & P_2 \\ P_4 & P_1 \end{bmatrix} = \mathbb{F}_{P_{21},P_{32}}\begin{bmatrix} P_2 & P_3 \\ P_1 & P_4 \end{bmatrix} = \mathbb{F}_{P_{21},P_{32}}\begin{bmatrix} P_4 & P_1 \\ P_3 & P_2 \end{bmatrix}. \tag{A.1}$$

We have the following relations. We write $\Delta_i \equiv \Delta_{P_i}$ etc. $c$ denotes as usual the central charge.

- Idempotency of $\mathbb{F}$:

$$\int_0^\infty \mathrm{d}P_{32}\, \mathbb{F}_{P_{21},P_{32}}\begin{bmatrix} P_3 & P_2 \\ P_4 & P_1 \end{bmatrix} \mathbb{F}_{P_{32},P_{21}'}\begin{bmatrix} P_4 & P_3 \\ P_1 & P_2 \end{bmatrix} = \delta(P_{21} - P_{21}'). \tag{A.2}$$

- Hexagon equation:

$$\int_0^\infty \mathrm{d}P_{32}\, \mathrm{e}^{\pi i(\sum_{i=1}^4 \Delta_i - \Delta_{21} - \Delta_{32} - \Delta_{31})} \mathbb{F}_{P_{21},P_{32}}\begin{bmatrix} P_3 & P_2 \\ P_4 & P_1 \end{bmatrix} \mathbb{F}_{P_{32},P_{31}}\begin{bmatrix} P_1 & P_3 \\ P_4 & P_2 \end{bmatrix}$$
$$= \mathbb{F}_{P_{21},P_{31}}\begin{bmatrix} P_3 & P_1 \\ P_4 & P_2 \end{bmatrix}. \tag{A.3}$$

- Pentagon equation:

$$\int_0^\infty \mathrm{d}P_{32}\, \mathbb{F}_{P_{21},P_{32}}\begin{bmatrix} P_3 & P_2 \\ P_{54} & P_1 \end{bmatrix} \mathbb{F}_{P_{54},P_{51}}\begin{bmatrix} P_4 & P_{32} \\ P_5 & P_1 \end{bmatrix} \mathbb{F}_{P_{32},P_{43}}\begin{bmatrix} P_4 & P_3 \\ P_{51} & P_2 \end{bmatrix}$$
$$= \mathbb{F}_{P_{21},P_{51}}\begin{bmatrix} P_{43} & P_2 \\ P_5 & P_1 \end{bmatrix} \mathbb{F}_{P_{54},P_{43}}\begin{bmatrix} P_4 & P_3 \\ P_5 & P_{21} \end{bmatrix}. \tag{A.4}$$

- SL$(2,\mathbb{Z})$ transformations:

$$\int_0^\infty \mathrm{d}P_2\, \mathbb{S}_{P_1,P_2}[P_0]\mathbb{S}_{P_2,P_3}[P_0] = \mathrm{e}^{\pi i \Delta_0}\delta(P_1 - P_3), \tag{A.5a}$$

$$\int_0^\infty \mathrm{d}P_2\, \mathbb{S}_{P_1,P_2}[P_0]\mathrm{e}^{-2\pi i \sum_{i=1}^3 (\Delta_i - \frac{c}{24})} \mathbb{S}_{P_2,P_3}[P_0] = \mathbb{S}_{P_1,P_3}[P_0]. \tag{A.5b}$$

- Relation at genus 1 and two punctures:

$$\mathbb{S}_{P_1,P_2}[P_3] \int \mathrm{d}P_4 \, \mathbb{F}_{P_3,P_4} \begin{bmatrix} P_2 & P_0' \\ P_2 & P_0 \end{bmatrix} e^{2\pi i(\Delta_4 - \Delta_2)} \mathbb{F}_{P_4,P_5} \begin{bmatrix} P_0 & P_0' \\ P_2 & P_2 \end{bmatrix}$$

$$= \int \mathrm{d}P_6 \, \mathbb{F}_{P_3,P_6} \begin{bmatrix} P_1 & P_0 \\ P_1 & P_0' \end{bmatrix} \mathbb{F}_{P_1,P_5} \begin{bmatrix} P_0 & P_0' \\ P_6 & P_6 \end{bmatrix} e^{\pi i(\Delta_0 + \Delta_0' - \Delta_5)} \mathbb{S}_{P_6,P_2}[P_5]. \quad \text{(A.6)}$$

They were written in this form in [51].[36]

# B Spacelike and timelike Liouville theory

In this appendix, we recall some known facts about both spacelike and timelike Liouville theory. The spacelike case is well-understood [42, 47, 48]. The timelike case is a bit more obscure, but essentially developed to the same amount of completeness and rigour from a bootstrap point of view [61, 62, 64–66]. We use conventions here that are particularly adapted to studying three-dimensional gravity.

## B.1 Conventions for spacelike Liouville theory

We will parametrize the central charge and conformal weights in the standard way,

$$c = 1 + 6Q^2, \qquad Q = b + b^{-1}, \qquad \Delta = \alpha(Q - \alpha), \qquad \alpha = \frac{Q}{2} + iP. \quad \text{(B.1)}$$

We will assume that $c \geq 25$ (and hence $b$ can be chosen in $b \in (0,1]$), but this assumption can be relaxed to $c > 1$. For states in the spectrum, we have $P \in \mathbb{R}$. We will denote vertex operators by $V_P(z)$. We use conventions such that the three-point function is given by

$$\langle V_{P_1}(0) V_{P_2}(1) V_{P_3}(\infty) \rangle = C_0(P_1, P_2, P_3), \quad \text{(B.2)}$$

where the structure constant $C_0$ is given by

$$C_0(P_1, P_2, P_3) = \frac{\Gamma_b(2Q)\Gamma_b(\frac{Q}{2} \pm iP_1 \pm iP_2 \pm iP_3)}{\sqrt{2}\Gamma_b(Q)^3 \prod_{k=1}^{3} \Gamma_b(Q \pm 2iP_k)}. \quad \text{(B.3)}$$

Here, a $\pm$ means that we are taking the product over all possible sign choices. For example, $\Gamma_b(\frac{Q}{2} \pm iP_1 \pm iP_2 \pm iP_3)$ denotes a product of eight terms. $\Gamma_b(z)$ is the double Gamma-function, which is explicitly defined through (2.19). $\Gamma_b(z)$ has simple poles for

$$z = -mb - nb^{-1}, \quad m, n \in \mathbb{Z}_{\geq 0}. \quad \text{(B.4)}$$

The structure constant (2.17) has simple poles[37] associated with double-twist operators [67, 159]

$$\alpha_i = \alpha_j + \alpha_k + mb + nb^{-1}, \quad m, n \in \mathbb{Z}_{\geq 0}, \quad \text{(B.5)}$$

and all reflections thereof, and simple zeros associated with degenerate representations of the Virasoro algebra

$$\alpha_i = -\frac{1}{2}(mb + nb^{-1}), \ Q + \frac{1}{2}(mb + nb^{-1}), \quad m, n \in \mathbb{Z}_{\geq 0}. \quad \text{(B.6)}$$

---

[36]There are two typos in [51] which we correct here. $\chi_b$ in eq. (6.34e) should be $-\frac{c}{24}$ and not as stated $\frac{c}{24}$. In eq. (6.34f), the product of Dehn twists should be $T_{\beta_4}^{-1} T_{\beta_2}$ and not $T_{\beta_4} T_{\beta_2}^{-1}$.

[37]For the poles to be simple we assume $b^2 \notin \mathbb{Q}$.

In this normalization, the reflection coefficient is unity and hence vertex operators are identified according to

$$V_P(z) = V_{-P}(z).$$ (B.7)

The two-point function is obtained by taking the limit $P_3 \to \frac{iQ}{2}$ in $C_0(P_1, P_2, P_3)$. Taking the limit is a bit subtle, but the result is

$$\langle V_P(0) V_{P'}(\infty) \rangle = \rho_0(P)^{-1} \delta(|P| - |P'|),$$ (B.8)

with

$$\rho_0(P) = 4\sqrt{2} \sinh(2\pi b P) \sinh(2\pi b^{-1} P).$$ (B.9)

which gives the inverse two-point function. This normalization of vertex operators is natural because $\rho_0(P)\, dP$ is the Plancherel measure on the Virasoro group (which is identical to the Plancherel measure on the modular double $U_q(\mathfrak{sl}(2, \mathbb{R}))$) [50, 160].

The conformal block expansion of an $n$-point function of local operators on a genus-$g$ Riemann surface in the Liouville CFT hence takes the form

$$\left\langle \prod_{i=1}^n V_{P_i}(z_i) \right\rangle_g = \int_0^\infty \prod_{a=1}^{3g-3+n} dP_a\, \rho_0(P_a) \prod_{\substack{(P_i, P_j, P_k) \\ \text{pair of pants}}} C_0(P_i, P_j, P_k) \left| \mathcal{F}_{g,n}^{\mathcal{C}}(\vec{P}) \right|^2.$$ (B.10)

Here we used the same notation for the conformal blocks as in the main text. They of course also depend on all the moduli of the problem, which we have suppressed on the right-hand side.

Let us pause to contrast this with the DOZZ formula for the structure constants of spacelike Liouville as conventionally written in the literature. It is given by [47, 48]

$$C_{\mathrm{DOZZ}}(P_1, P_2, P_3) = \frac{\tilde{\mu}^{-(\frac{Q}{2} + iP_1 + iP_2 + iP_3)} \Upsilon_b'(0) \prod_j \Upsilon_b(Q + 2iP_j)}{\Upsilon_b(\frac{Q}{2} + i\sum_j P_j) \prod_k \Upsilon_b(\frac{Q}{2} + i\sum_j P_j - 2iP_k)},$$ (B.11)

where

$$\tilde{\mu} := \left( \pi \mu \gamma(b^2) b^{2-2b^2} \right)^{1/b},$$ (B.12)

with $\gamma(z) = \Gamma(z)/\Gamma(1-z)$, and

$$\Upsilon_b(z) = \frac{1}{\Gamma_b(z)\Gamma_b(Q-z)}.$$ (B.13)

Here $\mu$ is the Liouville cosmological constant. The two-point functions inherited from the DOZZ formula are not canonically normalized

$$C_{\mathrm{DOZZ}}(P_1, P_2, \tfrac{iQ}{2}) = 2\pi \left[ \delta(P_1 + P_2) + S(P_1)\delta(P_1 - P_2) \right],$$ (B.14)

where $S(P)$ is the reflection amplitude

$$\tilde{\mu}^{-2iP} \frac{\Upsilon_b(Q + 2iP)}{\Upsilon_b(2iP)}.$$ (B.15)

Moreover the DOZZ formula is not invariant under reflection of the Liouville momenta; rather it picks up a factor of the reflection amplitude

$$C_{\mathrm{DOZZ}}(P_1, P_2, P_3) = S(P_1) C_{\mathrm{DOZZ}}(-P_1, P_2, P_3).$$ (B.16)

The universal formula $C_0$ can be written in terms of the DOZZ formula in the following way

$$C_0(P_1, P_2, P_3) = \left( \frac{\tilde{\mu}^{\frac{Q}{2}}}{2^{\frac{3}{4}}\pi} \frac{\Gamma_b(2Q)}{\Gamma_b(Q)} \right) \frac{C_{\text{DOZZ}}(P_1, P_2, P_3)}{\sqrt{\prod_{k=1}^{3} S(P_k)\rho_0(P_k)}} \,. \tag{B.17}$$

We prefer to work with $C_0$ rather than $C_{\text{DOZZ}}$ because it is reflection-symmetric (the conformal weights depend only on $P^2$ and so all CFT quantities ought to manifest reflection symmetry) and it eliminates the presence of the Liouville cosmological constant. Moreover the inverse of the two-point function inherited from $C_0$ admits a clean interpretation as the Plancherel measure of the Virasoro group.

## B.2 Conventions for timelike Liouville theory

Let us now move on to timelike Liouville. We will also parametrize it by the parameter $b$, but it is now related to the central charge and conformal weights as

$$\hat{c} = 1 - 6\hat{Q}^2 \,, \qquad \hat{Q} = b - b^{-1} \,, \qquad \hat{\Delta} = -\frac{\hat{Q}^2}{4} + \hat{P}^2 \,. \tag{B.18}$$

Hatted quantities denote quantities of the timelike theory. For the internal spectrum that appears in the operator product expansion, we have $\hat{P} \in \mathbb{R}$. However, external states in correlation functions can be analytically continued to any value $\hat{P} \in \mathbb{C}$ and we are mostly interested in $\hat{P} \in i\mathbb{R}$ [62, 63].

We will denote the vertex operators by $\hat{V}_{\hat{P}}(z)$. We use conventions in which the three-point functions take the form

$$\langle \hat{V}_{\hat{P}_1}(0)\hat{V}_{\hat{P}_2}(1)\hat{V}_{\hat{P}_3}(\infty)\rangle = \frac{1}{C_0(i\hat{P}_1, i\hat{P}_2, i\hat{P}_3)} \,. \tag{B.19}$$

Notice that the two-point function inherited from (B.19) (obtained by sending one of the hatted Liouville momenta to $\hat{Q}/2$) is non-diagonal. This is a signal that the timelike Liouville CFT contains a weight-zero operator that is distinct from the degenerate representation of the Virasoro algebra corresponding to the identity operator.

The conformal block expansion of a correlation function of local operators takes the form

$$\left\langle \prod_{i=1}^{n} \hat{V}_{\hat{P}_i}(z_i) \right\rangle = \int_{-\infty+i\varepsilon}^{\infty+i\varepsilon} \prod_{a=1}^{3g-3+n} \frac{d\hat{P}_a (\mathcal{N}\hat{P}_a^2)}{\rho_0(i\hat{P}_a)} \prod_{\substack{(P_i, P_j, P_k) \\ \text{pair of pants}}} C_0(i\hat{P}_i, i\hat{P}_j, i\hat{P}_k)^{-1} \left| \mathcal{F}_{g,n}^{\mathcal{C}}(\vec{\hat{P}}) \right|^2 \,. \tag{B.20}$$

The integration contour runs slightly above the real axis and avoids all the poles of the integrand. Shifting the contour by $+i\varepsilon$ avoids all the poles that the integrand has and gives a well-defined integrand. It was shown in [62] that this prescription indeed leads to crossing-symmetric correlation functions. The integration measure over $\hat{P}_a$ is fixed by the requirement of crossing symmetry. This only determines it up to an overall normalization $\mathcal{N}$. The normalization $\mathcal{N}$ essentially reflects the coupling constant of the theory. Our arguments do not fix the explicit value of $\mathcal{N}$ that we should choose such that the eq. (2.21) holds true.

## C  Some details about 3-manifold topology

Let us recall some geometry of hyperbolic 3-manifolds that are relevant to our study.

## C.1 Hyperbolic 3-manifolds via Kleinian groups

Let $M$ be a possibly non-compact connected complete hyperbolic 3-manifold. In three dimensions every hyperbolic manifold looks locally the same and hence the universal cover is the unique simply-connected hyperbolic manifold – $\mathbb{H}^3$. Consequently any hyperbolic 3-manifold can be obtained by a quotient $\mathbb{H}^3/\Gamma$, where $\Gamma \subset \mathrm{PSL}(2, \mathbb{C})$ is a discrete subgroup. Discreteness is needed in order to ensure that the action of $\Gamma$ on $\mathbb{H}^3$ is a properly discontinuous action. Such groups are known as Kleinian groups and there is an enormous body of mathematical literature on them and their connection with hyperbolic 3-manifolds [161, 162].

In order to get a regular hyperbolic 3-manifold, we also often want to assume that the Kleinian groups is torsion-free, i.e. does not contain elements $\gamma \in \Gamma$ with $\gamma^n = \mathbb{1}$ for some $n$. Such elements are elliptic and fix a line inside $\mathbb{H}^3$, thus leading to a conical deficit. Thus for a regular hyperbolic 3-manifold, $\Gamma$ does not contain any elliptic elements since if they are not finite order, the series $\{\gamma^n\}$ can get arbitrarily close to $\mathbb{1}$, which leads to a non-discrete group. Similarly parabolic elements in $\Gamma$ are associated to cusps in the 3-manifold. Thus to get a regular manifold, we also want to assume that we don't have any parabolic elements in $\Gamma$ and $\Gamma$ consists of purely loxodromic elements.

Since $\mathbb{H}^3$ is the universal cover of $M$, $\Gamma$ is naturally isomorphic to $\pi_1(M)$ (well-defined up to overall conjugation associated with the choice of base point). In fact, the isomorphism is given by the holonomy representation

$$\rho : \pi_1(M) \longrightarrow \Gamma . \tag{C.1}$$

Since $\mathbb{H}^3$ is contractible, we learn in particular that $\pi_2(M) = 0$ for any hyperbolic 3-manifold (as well as all higher homotopy groups). Such a space is called an Eilenberg MacLane space in topology and is often denoted by $K(\pi_1, 1)$. This means that any 2-sphere bounds a ball in $M$ and in particular every hyperbolic manifold is irreducible in the sense that it cannot be written as a connected sum of two smaller manifolds (except in the trivial way $M \# S^3$). Since Eilenberg MacLane spaces are determined up to homotopy equivalence, $\pi_1$ determines the manifold completely.

$\Gamma$ also acts on the boundary $\partial \mathbb{H}^3 = \mathbb{CP}^1$, but the action is not properly discontinuous. Instead, one has to remove a set $\Lambda$ (known as the limit set) from $\mathbb{CP}^1$ and $\Gamma$ acts properly discontinuously on the complement $\Omega = \mathbb{CP}^1 \setminus \Lambda$. One can thus extend $M$ to a manifold with boundary by setting

$$\overline{M} = (\mathbb{H}^3 \cup \Omega)/\Gamma . \tag{C.2}$$

$\Lambda$ is usually a very complicated set with non-trivial Haussdorff dimension. Note also that $\Omega$ can have many (possibly infinitely many) disconnected components and every component leads to a conformal boundary of $\overline{M}$ in the sense of the AdS/CFT correspondence.

There are a few cases in which $\Lambda$ is finite which lead to exceptions in many statements. They are known as elementary Kleinian groups and are convenient to exclude. In the torsion-free case without parabolic elements this only happens when $\pi_1$ is abelian and the corresponding 3-manifold is thermal $\mathrm{AdS}_3$ (i.e. a genus 1 handlebody). This is the only case where the torus without punctures can appear as a boundary component of a hyperbolic 3-manifold (with AdS boundary conditions). Indeed, let $\Omega_*$ be the component of $\Omega$ for which $\Omega/\Gamma \cong \mathbb{T}^2$ is a torus. Then $\Omega$ is a covering space of $\mathbb{T}^2$ (not necessarily the universal one) and as such $\Gamma$ is a subgroup of $\pi_1(\mathbb{T}^2) \cong \mathbb{Z}^2$. This means that $\Gamma$ is abelian and hence an elementary Kleinian group which only gives the case of the genus 1 handlebody.

There is also the important special case of a quasi-Fuchsian group in which the limiting set is topologically a $S^1 \subset S^2$ (although the $S^1$ can be a very fractal Jordan-curve). In this case, $\Omega$ consists of precisely two simply-connected regions. Let $\Omega_*$ be one of these regions. Since $\Omega_*$ is simply connected, it is a universal cover of the boundary surface. In particular by the same

argument as before, we learn that $\pi_1(M) \cong \pi_1(\Sigma_*)$, where $\Sigma_*$ is the corresponding boundary surface. Since both $M$ and $\Sigma_*$ are $K(\pi_1, 1)$'s, it follows that they are homotopy equivalent. In particular, the corresponding manifold $M$ is a Euclidean wormhole of topology $\Sigma \times [0, 1]$.

## C.2 Finiteness conditions

In order to state some of the results of 3-manifold topology, we have to assume some kind of finiteness condition on the 3-manifold under consideration. We will assume a somewhat strong finiteness condition that was also used in [163] of convex co-compactness. Let us first explain the definition of the convex core of a three-manifold. Define

$$C = \text{hyperbolic convex hull of } \Lambda \subset \mathbb{H}^3, \tag{C.3}$$

i.e. we first connect every pair of points in $\Lambda$ by a geodesic through the bulk and then take the convex hull of the resulting set of geodesics in the bulk. By construction, $C$ is preserved by the action of $\Gamma$. Thus we can form the convex core by taking the quotient

$$CM \equiv C/\Gamma. \tag{C.4}$$

By construction, also $\pi_1(CM) \cong \pi_1(M)$ and thus $CM$ is homotopy equivalent to $M$. When $\Gamma$ is "too simple", it can happen that the dimension of $CM$ is less than three. For example, for $\Gamma$ a quasi-Fuchsian wormhole, $CM$ is precisely the geodesic surface forming the throat of the wormhole.

$CM$ is now a hyperbolic 3-manifold (or 2-manifold) with boundary. The manifold is called convex co-compact when the convex core is compact (we also assume that $\Gamma$ is not elementary). The convex core is precisely the analogue of the bulk of the manifold that we consider in 2d JT gravity after we amputate the trumpets.

Convex co-compactness has many implications for other notions of finiteness:

1. $M$ is geometrically finite, which means that a slight thickening of $CM$ has finite volume:

$$(CM)_\delta = \{x \in \mathbb{H}^3 \mid d(x, CM) < \delta\}. \tag{C.5}$$

This is the most commonly used finiteness condition in the literature.

2. The previous point in particular makes it possible to have a finite renormalized volume of $M$, thus motivating the definition in the context of AdS/CFT.

3. The Dirichlet fundamental domain defined as

$$D_a = \{x \in \mathbb{H}^3 \mid d(x, a) < d(\gamma(x), a) \text{ for all } \gamma \neq \mathbb{1}\}, \tag{C.6}$$

is a polyhedron with finitely many sides.

4. $\Gamma$ is finitely generated.

5. $M$ has finitely many boundary components (this is the deep Ahlfors finiteness theorem).

Every of these properties alone is however usually weaker than convex co-compact.

### C.3  Rigidity

3-manifolds enjoy the remarkable property of being rigid. Colloquially speaking, the hyperbolic structure in the bulk is completely determined by the conformal structure of the boundary components, together with the topological type. To formalize this, it is useful to introduce two notions of deformation spaces of hyperbolic 3-manifolds. We suppress issues regarding compactifications of these deformation spaces.

Let us first define the moduli space of hyperbolic 3-manifolds with given fundamental group $\Gamma$. We have

$$\mathcal{M}_\Gamma = \{M \,|\, M \text{ hyperbolic, } \pi_1(M) \cong \Gamma\}/\sim, \tag{C.7}$$

where we identify isometric manifolds.

We also introduce $\text{Teich}(\Gamma)$, which will turn out to be the universal covering space of $\mathcal{M}_\Gamma$. Fix an element $M_0 \in \mathcal{M}_\Gamma$.

$$\text{Teich}(\Gamma) = \left\{ (M, f) \,\middle|\, \begin{array}{l} \text{M hyperbolic, } \pi_1(M) \cong \Gamma \\ f : M \longrightarrow M_0 \text{ an isomorphism} \end{array} \right\}\Big/\sim, \tag{C.8}$$

where $(M, f) \sim (M', f')$ if $M$ and $M'$ are isometric and $f \circ (f')^{-1}$ is homotopic to the identity. Intuitively, the Teichmüller space also keeps track of a marking. We can forget the information about $f$, which gives a covering map $\text{Teich}(\Gamma) \longrightarrow \mathcal{M}_\Gamma$. Then the most general rigidity result formulated by Bers, Maskit and Sullivan [80] states that

$$\text{Teich}(\Gamma) \cong \text{Teich}(\partial M), \tag{C.9}$$

i.e. the deformation space of the bulk precisely corresponds to the Teichmüller space of the boundary components. In some special cases, this theorem was known earlier, i.e. for the quasi-Fuchsian wormhole, this is known as Bers' uniformization theorem [164].

### C.4  Mapping class groups

We now carefully discuss how to take care of the mapping class groups. We want to explain the claim mentioned in section 2.7 that the three-dimensional mapping class group $\text{Map}_0(M, \partial M)$ that acts trivially on the boundary is trivial when boundaries are present.

Consider a mapping class group element $\gamma : M \longrightarrow M$. Because it acts trivially on the boundary and because of the isomorphism (C.9), we know that we can find again a representative $\gamma$ that is an isometry. Hence we can think of $\gamma$ as an $\text{PSL}(2, \mathbb{C})$ matrix that commutes with the action of the Kleinian group on $\mathbb{H}^3$.

But since we assumed that the boundary of $M$ is non-trivial, we know that the action of $\gamma$ on $\partial M$ and hence on all of $\partial \mathbb{H}^3$ is trivial. This implies that $\gamma$ is the identity and thus $\text{Map}_0(M, \partial M)$ is the trivial group.

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
