# Peer review of "Solving 3d Gravity with Virasoro TQFT"

_SciPost Physics, doi:SciPost Phys. 15, 151 (2023)_

## Round 2 · Referee Report · Anonymous (Referee 1) · 2023-7-15

Report

Dear Editor,

In this work, the authors perform the computation of 3d pure gravity amplitudes on hyperbolic 3-manifolds by constructing an explicit formalism in terms of Virasoro CFT. The main technical achievement is a prescription on how to deal with the mapping class group action as summarized in their equation (1.1), and how to decompose hyperbolic 3-manifolds in terms of computable building blocks. In particular, it shows us how to combine (local) TQFT techniques with the (non-local) large diffeomorphisms on the surface. 3d gravity has been attracting much attention lately, and the current work addresses and resolves an important problem.

As such, I highly recommend the paper for publication.

---

## Round 2 · Referee Report · Anonymous (Referee 2) · 2023-7-16

Strengths

  1. Largely solves an important open problem, namely how to compute amplitudes of 3d gravity on generic topologies.
  2. Very pedagogically written. Seemingly deep insights are gained from rather simple to follow calculations (the calculations are simple because the authors succeed in solving difficult problems in very elegant manners).
  3. Many new interesting technical results, for instance equation (1.1) and (2.21).

Report

This is an overall great piece of work. The authors solve a major open problem by formally solving the path integral of AdS3 gravity on arbitrary topologies. In many cases, the solution is not just formal as finite concrete amplitudes can be obtained. This can be considered a breakthrough.

The paper is very pedagogically written and was an absolute joy to read.

The fact that AdS3 gravity is closely related with the type of TQFT the authors discuss is not by itself new. To me, the most important key new element (though there are certainly others) consists of the steps between (2.61) and (2.63) (see also appendix C) where the authors explain that dividing out by the bulk mapping class group is compensated for hyperbolic 3 manifolds by part of the sum over bulk geometries (which includes the boundary mapping class group). This may sound like a technicality, but without this a solution can not be found. In some sense this is very similar to the reason why Mirzakhani recursion works for AdS2 gravity, if one goes through the proof.

Another piece of progress worth mentioning is equation (2.21), a very general result derived in a beatiful manner.

Lastly let me mention it was clever to use Heegaard splitting on hyperbolic 2 surfaces in order to cut and glue the three manifolds, since the more commonly used surgery on tori is ill defined for this theory.

I do have several small questions. 1. I was confused by a statement in section 2.1 that the metric is Lorentzian, whereas we are computing Eulidean path integrals. Perhaps the authors can explain this a bit more? 2. Between (2.26) and (2.27) the authors prove (2.21). They are an argument that string amplitudes decompose as Feynman diagrams. Do I understand correctly that for this argument to work it is key that we integrate over the whole Teichmuller space in (2.5)? Ordinarily as far as I understand closed string amplitudes do not have such a simple decomposition, because for closed strings we integrate only over the moduli space of Riemann surfaces (unlike for open strings where there is such a simple decomposition). If this is true, I think it would be worth clarifying this. If this is not true, additional explanation would also be helpful. 3. Why is the Hilbert space not spanned by conformal blocks of $SL(2)_k$? One would think this would be more natural from the bulk point of view, since ordinarly we think of the Hamiltonian reduction from $SL(2)_k$ to Virasoro as arising only on the boundary (because of the asymptotic boundary conditions). The quantum numbers might be identical, so perhaps the basis are related in a simple manner and the Virasoro blocks are just technically easier to work with?

Requested changes

  1. I would appreciate it if the authors could answer the above questions.
  2. In the mapping class group paragraph in the introduction when the statement is made that the bulk mapping class group is a subset of the boundary mapping class group, it would help to add a reference or refer to appendix C and section 2.7. At that point in the text it seems as if this is a fact that the reader is supposed to be aware of, and which is absolutely key in the derivation.
  3. In a note at the end of the introduction the authors suggest that by correctly giving people credit for their scientific contributions, we implicitly endorse their polotical convictions. I find this a dangerous precedent and believe SciPost should consider whether or not they want to back up such a view of how we value scientific contributions. With the ample use of Virasoro blocks (instead of $SL(2)_k$ blocks, see question 3 above) the name Virasoro TQFT is motivated enough.

  • validity: top
  • significance: top
  • originality: top
  • clarity: top
  • formatting: perfect
  • grammar: perfect

Author:  Lorenz Eberhardt  on 2023-07-20  [id 3825]

(in reply to Report 2 on 2023-07-16)
Category:
remark
objection

Dear referee,

We would like to comment on the 3rd requested change. We will respond to the rest of the scientific criticism once an editorial recommendation has been made.
The requested change misrepresents what we wrote in the paper and we fully agree that scientific contributions should be correctly attributed to their originator, regardless of politics. We however strongly feel that a person like O. Teichmuller does not deserve additional scientific honor for a recent development that he did not contribute to. In particular, this does not set a dangerous precedent for SciPost.
For reference, the relevant quote in our paper is ``We prefer the name “Virasoro TQFT” because we feel it better captures the physical meaning of the TQFT, and avoids the implicit endorsement of the political convictions of O. Teichmuller’’.

With best wishes

The authors

---

## Round 3 · Author Response

We would like to thank both referees for their careful reading of the manuscript and their feedback. Let us answer the excellent questions by referee 2:
-
Indeed, it is very important to start in Lorentzian signature, since the correct phase space can only be determined in Lorentzian signature. Starting with gravity in Euclidean signature would have led to the incorrect assertion that the phase space is given by (a subset of) the moduli space of flat $\mathrm{SL}(2,\mathbb{C}$ connections on the initial value surface. One can then proceed by quantizing the phase space to obtain the Hilbert space. Once one has determined the Hilbert space, one can switch to Euclidean signature, since essentially by definition, the Hilbert space of the theory does not care about the spacetime signature. In particular, the \emph{Euclidean} gravity partition function on $\Sigma \times \mathrm{S}^1$ can be obtained by tracing over the Hilbert space. We have added some comments along these lines at the beginning of Section 2.7.
-
Yes, it is correct that it is crucial that one integrates over all of Teichmuller space. The main reason is that Teichmuller space naturally factorizes as follows. Choosing a pair of pants decomposition of the surface, Teichmuller space can be globally parametrized by Fenchel-Nielsen coordinates associated to the decomposition. Thus in spacetime, the 'amplitude' corresponding to the inner product can be computed by a single 'Feynman-diagram' corresponding to the pair of pants decomposition. As mentioned by the referee, such a statement would be very wrong for actual string amplitudes, which instead unify all channels in one diagram. We added similar explanations in Section 2.3.
-
The answer to this question is very interesting. Teichmuller space can also be quantized by certain $\mathrm{PSL}(2,\mathbb{R})_k$ conformal blocks (although this has never been made precise to our knowledge). Only some conformal blocks appear since only a part of the moduli space of flat $\mathrm{PSL}(2,\mathbb{R})$ connections was quantized. There is a correspondence between correlation functions of the $\mathrm{PSL}(2,\mathbb{R})_k$ WZW model and Liouville theory, which also implies a simple relationship between the conformal blocks (see hep-th/0507114, equation 3.29 for $r=n-2$ for the genus 0 case). In particular, this implies that the crossing properties of the $\mathrm{PSL}(2,\mathbb{R})_k$ blocks are equivalent and thus the Hilbert space obtained from quantization in terms of $\mathrm{PSL}(2,\mathbb{R})_k$ blocks is equivalent. From the point of view of quantization, whether one obtains current algebra blocks or Virasoro blocks depends on a choice of polarization (i.e. coordinates on which the wave function depends). In the usual quantization known from Chern-Simons theory, the wavefunction depends homomorphically on the gauge connection, i.e. is a "function" of $A^a_z$ for $a=3,+,-$ (it is actually a holomorphic section of a certain line bundle). To obtain Virasoro conformal blocks, one has to use the polarization described in 10.1016/0550-3213(90)90510-K. The fact that they lead to the same mapping class group representations, means that the two quantizations are equivalent. In particular, we could have also taken the quantization in terms of $\mathrm{PSL}(2,\mathbb{R})_k$ blocks; for the computation of partition functions on closed manifolds this gives on the nose the same result. In the presence of boundaries, the partition function takes values in $\mathrm{PSL}(2,\mathbb{R})_k$ blocks, to get the correct answer one has to implement Hamiltonian reduction which is the aforementioned connection to Virasoro conformal blocks. For the purpose of AdS/CFT, it is natural and simpler to directly use the quantization in terms of Virasoro conformal blocks. Some of us plan to return to these issues in a future publication.

---

## Round 3 · List of Changes

1. See the comments above.
2. Good point, we implemented this in the new version.
3. We changed the statement to 'We prefer the name “Virasoro TQFT” because we feel it better captures the physical meaning of the TQFT, and we strongly feel that a person like O. Teichmüller does not deserve additional scientific honor for a recent development that he did not contribute to'. This should make it impossible to misinterpret the statement.

You are currently on this page

---

## Editorial Decision

published